# Arguments for the biological and predictive relevance of the proportional recovery rule

Jeff Goldsmith[1]*, Tomoko Kitago[2,3], Angel Garcia de la Garza[1], Robinson Kundert[4], Andreas Luft[4,5], Cathy Stinear[6], Winston D Byblow[7], Gert Kwakkel[8,9], John W Krakauer[10,11,12,13]

[1]Department of Biostatistics, Columbia Mailman School of Public Health, New York, United States; [2]Burke Neurological Institute, White Plains, United States; [3]Weill Cornell Medicine, New York, United States; [4]Cereneo, Center for Neurology and Rehabilitation, Vitznau, Switzerland; [5]Vascular Neurology and Neurorehabilitation, Department of Neurology, University Hospital Zurich, University of Zurich, Zurich, Switzerland; [6]Department of Medicine, University of Auckland, Auckland, New Zealand; [7]Department of Exercise Sciences, University of Auckland, Auckland, New Zealand; [8]Rehabilitation Research Centre, Reade, Amsterdam, Netherlands; [9]Rehabilitation Medicine, Amsterdam UMC - Location VUMC, Amsterdam Movement Sciences, Amsterdam, Netherlands; [10]Department of Neurology, Johns Hopkins University, Baltimore, United States; [11]Department of Neuroscience, Johns Hopkins University, Baltimore, United States; [12]Department of Physical Medicine and Rehabilitation, Baltimore, United States; [13]Santa Fe Institute, Santa Fe NM, United States

*For correspondence: ajg2202@cumc.columbia.edu

Competing interest: The authors declare that no competing interests exist.

**Abstract** The proportional recovery rule (PRR) posits that most stroke survivors can expect to reduce a fixed proportion of their motor impairment. As a statistical model, the PRR explicitly relates change scores to baseline values – an approach that arises in many scientific domains but has the potential to introduce artifacts and flawed conclusions. We describe approaches that can assess associations between baseline and changes from baseline while avoiding artifacts due either to mathematical coupling or to regression to the mean. We also describe methods that can compare different biological models of recovery. Across several real datasets in stroke recovery, we find evidence for non-artifactual associations between baseline and change, and support for the PRR compared to alternative models. We also introduce a statistical perspective that can be used to assess future models. We conclude that the PRR remains a biologically relevant model of stroke recovery.

## Editor's evaluation

This fundamental work provides a comprehensive look at validity of the Proportional Recovery Rule, which states that patients will recover a fixed proportion of lost function after stroke. By undertaking a thorough investigation of the statistical properties of the analysis of change and baseline values the authors elucidate the statistical framework that can be used regardless of the topic of study. In a compelling model comparison across several large sets of data, the authors confirm support for the Proportional Recovery Rule over other models of recovery.

## Introduction

Intuition, experience, and data suggest that patients with worse motor impairment in the immediate post-stroke period will also typically see the largest absolute reductions in impairment during the first 3–6 months of recovery. However, rigorously quantifying this observation has proved challenging. The proportional recovery rule (PRR) was an early attempt to describe the relationship between initial impairment and recovery through the investigation of upper-extremity Fugl-Meyer assessments (FMA-UE) at baseline and at subsequent follow-up visits, with recovery defined as the change over time (*Prabhakaran et al., 2008*; *Krakauer and Marshall, 2015*). This work indicated that, on average, a large subset of patients recovered roughly 70% of the maximal potential recovery from impairment, but a biologically distinct and smaller subgroup recovered much less ('non-recoverers'). Since its introduction, the PRR has been applied across several neurological domains, implemented in various ways across studies, and evaluated using several different statistical metrics (*Lazar et al., 2010*; *Winters et al., 2015*; *Winters et al., 2017*; *Veerbeek et al., 2018*; *Byblow et al., 2015*). In a recent article, we sought to reestablish the original conceptualization of the PRR as a regression model for describing of recovery from upper limb impairment (*Kundert et al., 2019*).

The PRR has, in recent years, come under fire. The criticisms echo concerns raised anytime change as an outcome is related to the baseline value, particularly before and after an intervention – whether the outcome of interest is blood pressure, CD4 cell count, or behavior. The concern hinges on statistical questions about the relationship between baseline values and change scores – a relationship that is often fraught, counterintuitive, and has confounded researchers and statisticians for decades (*Oldham, 1962*; *Gill et al., 1985*; *Tu, 2016*). Bringing these issues to the fore and illustrating the ways they affect the specific case of stroke recovery has spurred an important and informative debate. Specifically, there are questions about the usefulness of correlations in the case when there is mathematical coupling from inclusion of the baseline value in the change score, the distinction between population-level descriptions and patient-level predictions; the usefulness of the PRR in functional domains other than upper-extremity motor control; the identification of 'non-recoverers', both prospectively and retrospectively; and the possibility that ceiling effects exaggerate associations or introduce non-linearity not accounted for by the PRR (*Hope et al., 2018*; *Hawe et al., 2018*; *Bonkhoff et al., 2020*; *Bowman et al., 2021*).

Although the critiques of the PRR have been rigorous and careful, never asserting that the PRR was entirely irrelevant for recovery, the tone and content of some of this work might make it understandable for readers to infer otherwise. In 'Recovery after stroke: not so proportional after all?' (*Hope et al., 2018*), the authors say in their discussion that they 'are not claiming that the proportional recovery rule is wrong', only that they do not see evidence that the rule holds and that other models for recovery are possible. *Hawe et al., 2018*, included some similar caveats in the discussion of a paper titled 'Taking proportional out of recovery'. *Senesh and Reinkensmeyer, 2019* (title: 'Breaking proportional recovery after stroke') summarized these two papers in their abstract by saying they 'explained that [proportional recovery] should be expected because of mathematical coupling between the baseline and change score'. Some of the authors of the original criticisms have since moderated their stance and important points of consensus are emerging; see, for example, *Bonkhoff et al., 2020*, and *Bonkhoff et al., 2022*. Against this rapidly evolving and potentially confusing backdrop, we hope to bring together all the thorny issues so that readers can reach some degree of closure on the PRR.

The growing meta-literature often discusses the PRR in an abstract way, focusing on a baseline $x$, a single follow-up $y$, the change $\delta = y - x$, and all the various correlations among them. Readers could be forgiven for asking how an intuitive formula for expected recovery spawned its own cottage industry, and why arguments about the PRR have become so esoteric. We suspect that many, by now, would prefer a simple judgment on the truth of the PRR without a winding statistical detour. We're sympathetic to this perspective, but find the detour necessary as the nuanced and sometimes counterintuitive statistical arguments are critical to get right for the sake of furthering our understanding of the biological mechanisms of recovery.

Our goals are to synthesize and discuss the statistical issues relevant to the PRR clearly, to describe appropriate analysis techniques, to identify areas of emerging consensus, and to resolve arguments where possible. We discuss valid but non-standard hypothesis tests for correlations, framed to distinguish true signal from artifact. This is not the same as distinguishing between the PRR and other recovery mechanisms that induce strong correlations; we therefore evaluate competing models based

on their ability to predict outcomes. Much of this discussion assumes that 'recoverers' and 'non-recoverers' can be differentiated, and mainly focuses on models for recoverers. That said, the existence of distinct recovery groups complicates the quantification of recovery. We discuss the hazards of applying advanced statistical methods in settings where they aren't justified by the data, and discuss ways of testing whether observed data are consistent with a hypothesized mechanism. Throughout, we use generated datasets and data taken from several published studies of recovery to illustrate our statistical points. In some places, the discussion is unavoidably technical, but the aim is to focus on the details that are pertinent to the core question: Is there a true systematic relationship between impairment and change in impairment after stroke? If the answer is yes, then this would imply that there is important biological work to be done to explain the mechanism for this phenomenological regularity.

## Results

### Limitations of the correlation between baseline and change as a measure of association

Although the PRR is best understood as a regression model (*Kundert et al., 2019*), correlations between baseline and follow-up, and between baseline and change, have often been used to summarize data and are presented as evidence for the rule. A focus on $\text{cor}(x, \delta)$, where $\delta = y - x$ is the change between follow-up ($y$) and baseline ($x$), is intuitive in the context of recovery. The value of this correlation can be affected in unexpected ways due to mathematical coupling, most broadly defined as the setting where one value ($x$) is also included in the definition of the second ($\delta$). The following relationship is known to hold:

$$\text{cor}(x, \delta) = \frac{\sigma_y \text{cor}(x,y) - \sigma_x}{\sqrt{\sigma_y^2 + \sigma_x^2 - 2\sigma_x\sigma_y\text{cor}(x,y)}} = \frac{\sqrt{k}\text{cor}(x,y) - 1}{\sqrt{1 + k - 2\sqrt{k}\text{cor}(x,y)}} \quad (1)$$

This equation shows the dependence of $\text{cor}(x, \delta)$ on $\text{cor}(x, y)$ and the variance ratio $k = \frac{\sigma_y^2}{\sigma_x^2}$. The relationship between these quantities, visualized first by *Bartko and Pettigrew, 1968*, and more recently as a three-dimensional surface by *Hope et al., 2018*, is shown as a contour plot in *Figure 1*. For reasons that will become clear shortly, we highlight the contour corresponding to $k = 1$, where baseline and follow-up have equal variance. Other contour lines in this figure correspond to fixed values of $k$ ranging between 0.01 and 4.

An immediate observation from this plot is that the range of possible values for $\text{cor}(x, \delta)$ depends on $\text{cor}(x, y)$. When $\text{cor}(x, y) = 0$, for example, $\text{cor}(x, \delta)$ is restricted to lie in [-1,0] rather than [-1,1]. By itself, this suggests that usual tests for the significance of a correlation in which the null value is

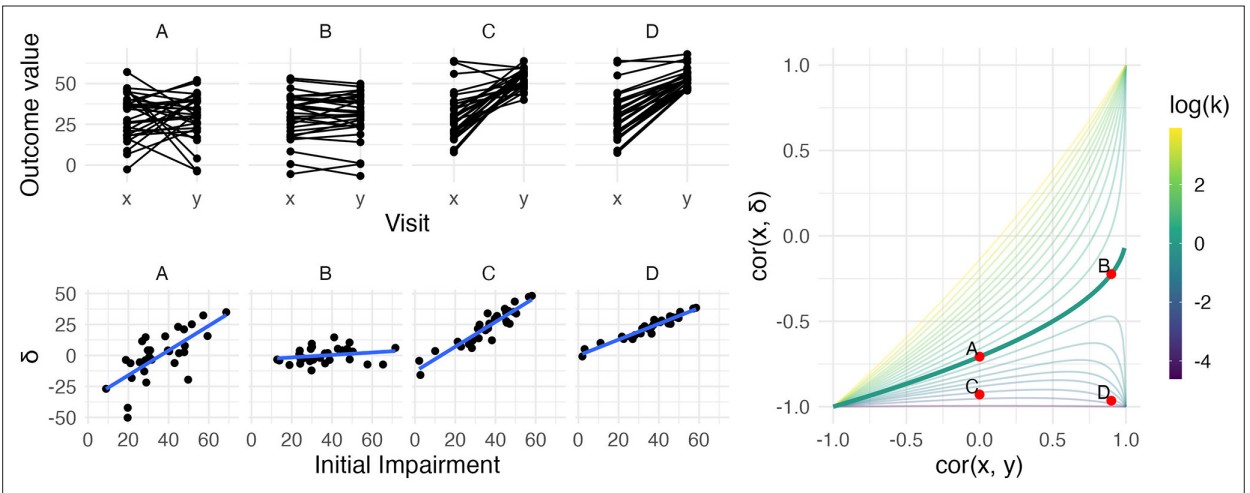

**Figure 1.** The left panels show four generated datasets, labeled A, B, C, and D. In the top row, panels show outcome values and baseline (*x*) and follow-up (*y*). In the bottom row, panels show change (*δ*) against initial impairment (66 − *x*). The right panel shows a contour plot of *Equation 1*, with contours corresponding to values of the variance ratio *k* and the contour for *k*=1 highlighted. Points on this surface show correlation values obtained for Datasets A through D. A figure similar to the contour plot is shown in *Hope et al., 2018*, with different axes and orientations.

assumed to be 0 are inappropriate for investigations of recovery. However, as we'll see shortly, this does not mean that hypothesis tests are impossible – only that more appropriate ones are necessary.

The dependence of $\mathrm{cor}\,(x, \delta)$ on $\mathrm{cor}\,(x, y)$ and $k = \frac{\sigma_y^2}{\sigma_x^2}$ is the basis for two related criticisms of using the correlation between baseline and change as a statistical measure of recovery. First, the canonical example of coupling is the setting in which baseline $(x)$ and follow-up $(y)$ are uncorrelated and have the same variance. This situation is represented by the point on the surface where $\mathrm{cor}\,(x, y) = 0$ and the variance ratio $k = \frac{\sigma_y^2}{\sigma_x^2} = 1$. In this setting $\mathrm{cor}\,(x, \delta) = -0.71$ – a value that, for some, is unexpectedly high and casts doubt on any large correlation between baseline and change (*Hope et al., 2018*; *Hawe et al., 2018*).

A broader argument relates to settings where measurements at follow-up have much lower variance than initial values, as is typically the case for studies of recovery. In cases where $k$ is small, $\mathrm{cor}\,(x, \delta)$ may be '(non-trivially) stronger than $\mathrm{cor}\,(x, y)$' and therefore spurious or misleading (*Hope et al., 2018*). Settings with small values of $k$ have been described as 'degenerate' in that $\mathrm{cor}\,(x, \delta)$ will approach $-1$ (*Bowman et al., 2021*); as *Figure 1* makes clear, this is a concern for any value of $\mathrm{cor}\,(x, y)$.

The recognition of these statistical issues, and the role they have played in understanding recovery, reveals some limitations of using correlation to measure the association between baseline and change and produce evidence for the significance of this association. By itself, $\mathrm{cor}\,(x, \delta)$ will give at best an incomplete understanding of recovery, and traditional hypothesis tests (focusing on 0 as a null value) are inappropriate. That said, these criticisms don't invalidate the PRR – they aren't even directly relevant to the PRR. Instead, they clarify the importance of understanding the relationship between $\mathrm{cor}\,(x, \delta)$, $\mathrm{cor}\,(x, y)$, and the variance ratio $k = \frac{\sigma_y^2}{\sigma_x^2}$; the importance of each these in the studies of recovery; and the use of appropriate summaries of the data.

## Distinguishing true and artifactual signals

To paraphrase the previous section, when the variance ratio is small, large values of $\mathrm{cor}\,(x, \delta)$ can arise from a wide range of $\mathrm{cor}\,(x, y)$. It has been argued that high correlations between baseline and change are invalid unless they are accompanied by high correlations between baseline and follow-up.

Refutation of this view has a long history in the statistical literature. *Oldham, 1962*, argues that a variance ratio other than 1 is evidence for some real effect or process: 'Unless some agent has caused a change of standard deviation between the two occasions, $\sigma_x^2$ will equal $\sigma_y^2$'. This argument is slightly more complicated when the outcome measure is bounded, a setting Oldham did not consider but that arises in stroke recovery. Heterogeneous recovery that depends on initial impairment could be the agent that causes a reduction in variance; alternatively, variance may be reduced because recovery is homogeneous but subject to ceiling effects or because the impairment scale is non-linear. In any case, when $k < 1$ the amount of recovery is related to the baseline value in a way that is not attributable to mathematical coupling, and differentiating between explanations is necessary after ruling out coupling. Oldham's method, which derives from the role of the variance ratio in studies where baseline and change are important, formalizes this concept and is a commonly used approach for understanding the relationship between baseline and change; it will be presented in the next section.

The value of $\mathrm{cor}\,(x, \delta)$ depends on the variance ratio $k$ and on $\mathrm{cor}\,(x, y)$. The variance ratio can be used as a measure of the extent to which $\delta$ depends on baseline values, regardless of the value of $\mathrm{cor}\,(x, y)$, with the understanding that this dependence can arise from the PRR or other recovery mechanisms. Meanwhile, $\mathrm{cor}\,(x, y)$ indicates whether follow-up values are related to baseline, regardless of $k$. Thus $\mathrm{cor}\,(x, y)$ is relevant for questions of patient-level prediction although, as we'll argue later, correlations are less useful than direct measures of prediction accuracy.

We simulate four datasets with different values of $\mathrm{cor}\,(x, y)$ and $k$:

- A: $\mathrm{cor}\,(x, y) = 0$ and $k = 1$
- B: $\mathrm{cor}\,(x, y) = 0.9$ and $k = 1$
- C: $\mathrm{cor}\,(x, y) = 0$ and $k = 0.16$
- D: $\mathrm{cor}\,(x, y) = 0.9$ and $k = 0.16$

All datasets have $x$ values generated from a Normal distribution with mean 30 and standard deviation 14, and consist of 30 generated subjects. Datasets A and B have follow-up values $(y)$ with mean 30, while Datasets C and D have follow-up values with mean 53 (variances at follow-up are determined

by the variance ratio). Generated datasets did not include values below 0 or above 66. These datasets were generated to clarify the relationship between $\text{cor}\,(x, y)$, $k$, and $\text{cor}\,(x, \delta)$: they are not assumed to follow the PRR or any other explicit model for recovery. The left panels in **Figure 1** show baseline and follow-up values (top row) and scatterplots of $\delta$ against initial impairment (bottom row), with initial impairment defined as $\text{FM}_{\max} - x = 66 - x$. The right panel indicates the placement of these datasets on the contour plot of $\text{cor}\,(x, \delta)$.

Dataset A is a canonical example of mathematical coupling, a setting that results in $\text{cor}\,(x, \delta) = -0.71$. Like Dataset A, Dataset C has $\text{cor}\,(x, y) = 0$, but the variance ratio is lower. Given our emphasis on $k$ and on $\text{cor}\,(x, y)$, any debate about whether $\text{cor}\,(x, \delta)$ for this dataset might be called 'spurious' is less relevant than (i) the true reduction in variance that results from recovery and (ii) the true association between the baseline value and the magnitude of change. Indeed, despite the inability of the baseline value to usefully predict follow-up in Dataset C, these data represent a case in which baseline values can be used to predict change in a non-artifactual way, which is a setting that Oldham and many others since have argued is important (**Oldham, 1962**; **Tu and Gilthorpe, 2007**).

In contrast, Datasets B and D have large $\text{cor}\,(x, y)$, which suggests an ability to predict follow-up using baseline with some degree of accuracy. In Dataset B, change from baseline to follow-up is constant with some patient-level noise, and accurate predictions at follow-up are a simple byproduct of that constancy. Dataset B is also, arguably, irrelevant: whether due to measurements that are truly non-linear, ceiling effects, or proportionality, motor impairment recovery is marked by heterogenous recovery across subjects.

Dataset D represents the least controversial scenario among our generated data: there is recovery heterogeneity that is predictable based on initial values and results in a variance ratio that is less than one, and the initial values meaningfully predict outcomes at follow-up. The variance ratio and correlations don't arise either from mathematical coupling or from regression to the mean due to measurement error. Real data that are similar to these are suggestive of an underlying biological recovery process in which baseline values predict change and final outcomes.

Taken together, the generated datasets in this section illustrate the relationship between $\text{cor}\,(x, y)$, $k$, and $\text{cor}\,(x, \delta)$, as well as the kinds of observed data that can give rise to various combinations of these values. The examples highlight discrepancies between $\text{cor}\,(x, y)$ and $\text{cor}\,(x, \delta)$ to illustrate their shortcomings when viewed individually. We also identify a setting, typified by Dataset D, in which each measure suggests the presence of a relevant association. However, it is not the case that data like these necessarily imply that recovery follows the PRR. Other biological models, the presence of strong ceiling effects, or other mechanisms could produce data similar to Dataset D, and how to compare competing models will be considered in later sections.

## Recasting Oldham's method

*Equation (1)* and *Figure 1* show that $\text{cor}\,(x, y)$ and $k$ determine the value of $\text{cor}\,(x, \delta)$ in ways that can be counterintuitive. When $\text{cor}\,(x, y) = 0$, for example, an appropriate null value for a hypothesis test of $\text{cor}\,(x, \delta)$ is –0.71 rather than 0, because this corresponds to 'no recovery' or $k = 1$; a table in the appendix provides null values for hypothesis tests of $\text{cor}\,(x, \delta)$ under a range of values of $\text{cor}\,(x, y)$. However, like others, we think the possible confusion around $\text{cor}\,(x, \delta)$ as a measure of evidence make it less suitable than other approaches.

Oldham's method suggests to use $\text{cor}\,\left(\frac{x+y}{2}, x - y\right)$, or simply $\text{cor}\,(x + y, x - y)$, in place of $\text{cor}\,(x, \delta)$. This correlation is zero, rather than −0.71, in the canonical example of mathematical coupling. Indeed, this correlation is zero if and only if $k = 1$ – regardless of $\text{cor}\,(x, y)$, and even in many cases where measurement error or other processes might affect the ability to reliably measure outcomes at baseline and follow-up. Thus, Oldham's method often guards against false conclusions due to mathematical coupling and regression to the mean due to measurement error.

Instead of $\text{cor}\,(x + y, x - y)$, we prefer to focus on the variance ratio $k$. Values of $k$ that differ from 1 suggest associations between baseline and change that do not arise from mathematical coupling or regression to the mean. In parallel, we examine the correlation $\text{cor}\,(x, y)$, which is equal to zero under the null hypothesis that follow-up values are uncorrelated with baseline. These are important but distinct; a suggestion that very small values of $k$ can only produce spurious correlations $\text{cor}\,(x, y)$ would stem from conflating the two. We again emphasize that these tests are intended to assess whether correlations are 'artifactual' (arising from mathematical coupling or regression to the mean), but not

to evaluate support for the PRR in comparison to competing models. For instance, as noted by both (*Hope et al., 2018* and *Hawe et al., 2018*), Oldham's method does not address the possibility of ceiling effects; in our view, determining which process gives rise to observed correlations comes after assessing whether those correlations are artifacts driven by coupling.

Parametric hypothesis tests are available for both $k$ and $\mathrm{cor}(x, y)$, but depend on assumptions that may be unmet in the context of stroke recovery. We also argue for specific null values of both $k$ and $\mathrm{cor}(x, y)$, although other choices are possible; one might instead choose 'random recovery' described in *Lohse et al., 2021*, as a null distribution and identify corresponding values. Instead of a parametric or simulation-based approach (*Tu et al., 2005*), we suggest a resampling-based one that can compare to the null values we suggest, and we describe how this method can be used for other null distributions. See Methods for details. A web-based app is available to carry out this analysis (https://jeff-goldsmith.shinyapps.io/prr_dashboard/).

The statistical significance of the recovery proportion and $R^2$ values have often been used as evidence for the PRR. Appendix 1 provides a detailed description of the connections between these and $\mathrm{cor}(x, \delta)$, $k$, and $\mathrm{cor}(x, y)$. In short, null-value hypothesis testing for the recovery proportion and the interpretation of $R^2$ values can be difficult for reasons that are analogous to those for correlations: appropriate null values are available but non-standard, and the range of possible values for $R^2$ can be limited. We urge caution in interpreting these quantities and so prefer to use other measures of association.

## Comparing distinct biological models for recovery

The PRR was developed as a model to explain recovery from motor impairment after stroke. Quantitative results, illustrated in Dataset C, show it's possible for recovery, defined as the change from baseline to follow-up, to be related to baseline values even when follow-up is not predicted well by baseline. But the ability to predict patient outcomes at follow-up is important in itself, and predictive performance can form a basis for comparing the PRR to alternative biological models for recovery. We

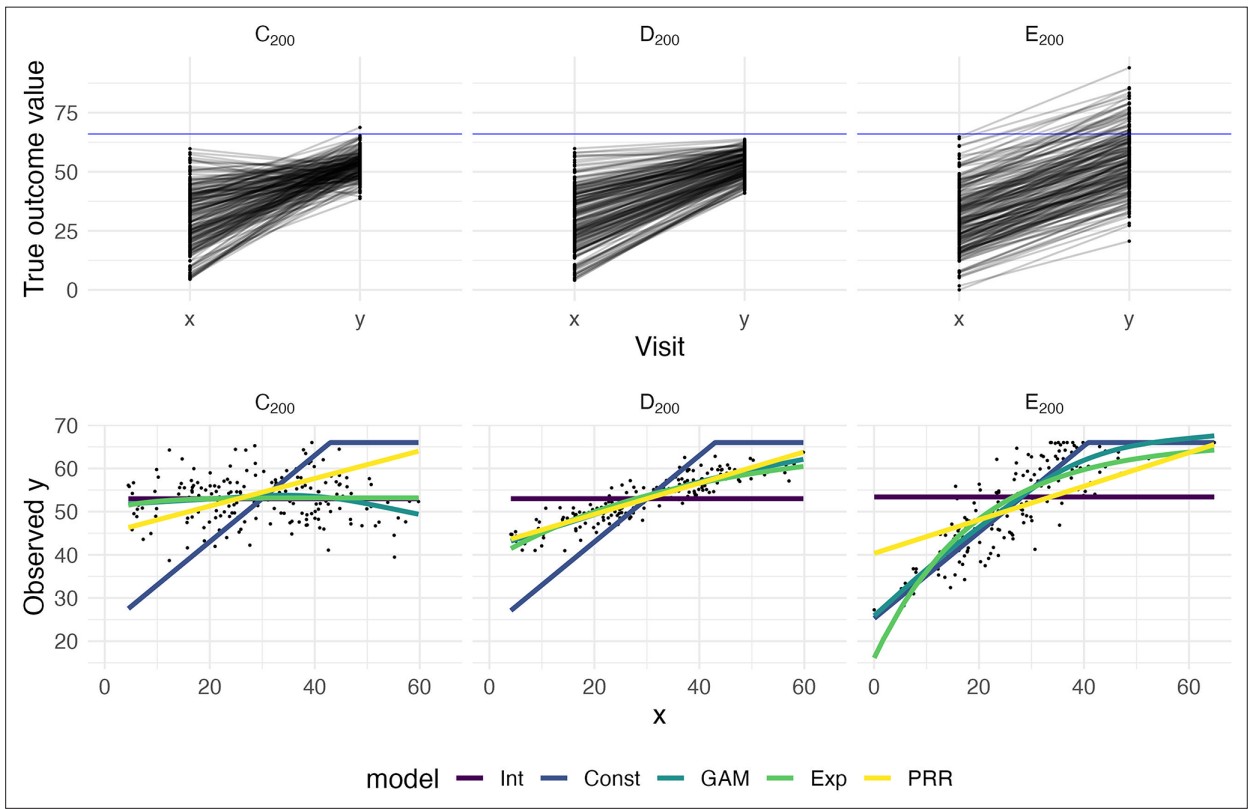

**Figure 2.** The top panels show three generated datasets with true outcome values and baseline (*x*) and follow-up (*y*); a horizontal line indicates a ceiling on observed values. In the bottom row, panels show the observed (ceiled) value at follow-up against the baseline value. Fitted values from an intercept-only model (Int), a generalized additive model (GAM), and the proportional recovery rule (PRR) are shown in the bottom row.

suggest cross validation (CV) as a means to compare models, using median absolute prediction error (MAPE) in held-out data to measure predictive accuracy (lower MAPE values reflect higher prediction accuracy). Details are available in Methods.

To illustrate how prediction accuracy might be used to distinguish between competing models for recovery, we generate three datasets under different recovery mechanisms and use CV to evaluate candidate models. The first two datasets, which we'll label $C_{200}$ and $D_{200}$, are generated using the same process as Datasets C and D above but consist of 200 patients. Dataset $E_{200}$ is similar to Dataset B, but has a latent follow-up mean of 70 rather than 30. Follow-up values greater than 66 are subject to strict threshold. That is, Dataset $E_{200}$ implements large, constant recovery (with some noise) and imposes a ceiling.

We consider several models for the association between $y$ and $x$. First, we assume that $y$ values are randomly distributed around a common mean value, and do not depend on $x$; this is an intercept-only regression model, and the mean of observed $y$ values is used to predict future outcomes. We implement constant recovery with a ceiling effect as a special case of Tobit regression, assuming $y = \min\{x + c + \epsilon, \, 66\}$, where the only free parameter is the constant $c$. This uses all data points, including those at ceiling, when estimating model parameters. We adopt an exponential form for recovery, motivated by **van der Vliet et al., 2020**, which includes an explicit ceiling. Next, we assume that $y$ depends on $x$, but allow the association to be smooth and non-linear using an additive model (generalized additive model [GAM]); details of this model are given in Appendix 1. Finally, we implement the PRR to estimate $\delta$ given $x$, with $y$ taken to be $x + \delta$. In all cases, we use available data to estimate model parameters. Note that except for the PRR, models focus on predicting the follow-up value directly. Also, the additive model includes several other models as special cases, but may overfit due to its flexibility.

The top row in **Figure 2** shows baseline and follow-up values, and the bottom row shows scatterplots of observed (ceiled) $y$ against $x$. The panels in the bottom row of **Figure 2** clarify the relationship between the data generating mechanism and the ability of $x$ to predict $y$; in each panel, fitted values from each of the candidate models are overlaid. Unsurprisingly, for Dataset $C_{200}$, there is no apparent relationship between $x$ and $y$ – these data are generated under $\mathrm{cor}\,(x, y) = 0$. Dataset $D_{200}$ is a case when baseline values are predictive of change *and* outcomes. In Dataset $E_{200}$, the expected $y$ increases linearly with $x$ until reaching a plateau, and then is uniformly near the maximum value.

**Figure 3** shows the result of our CV procedure applied to Datasets $C_{200}$, $D_{200}$, and $E_{200}$; panels show the distribution of MAPE across repeated training/testing splits. For Dataset $C_{200}$, the intercept-only, exponential, and additive models are the best performers; the PRR suffers from the lack of an intercept, which produces a bias in the predictions, while constant recovery with a ceiling is not well suited to this mechanism. For Dataset $D_{200}$, the intercept-only and constant recovery models are the worst performers, while the (true) PRR, exponential, and additive models are similar. Finally, for Dataset $E_{200}$,

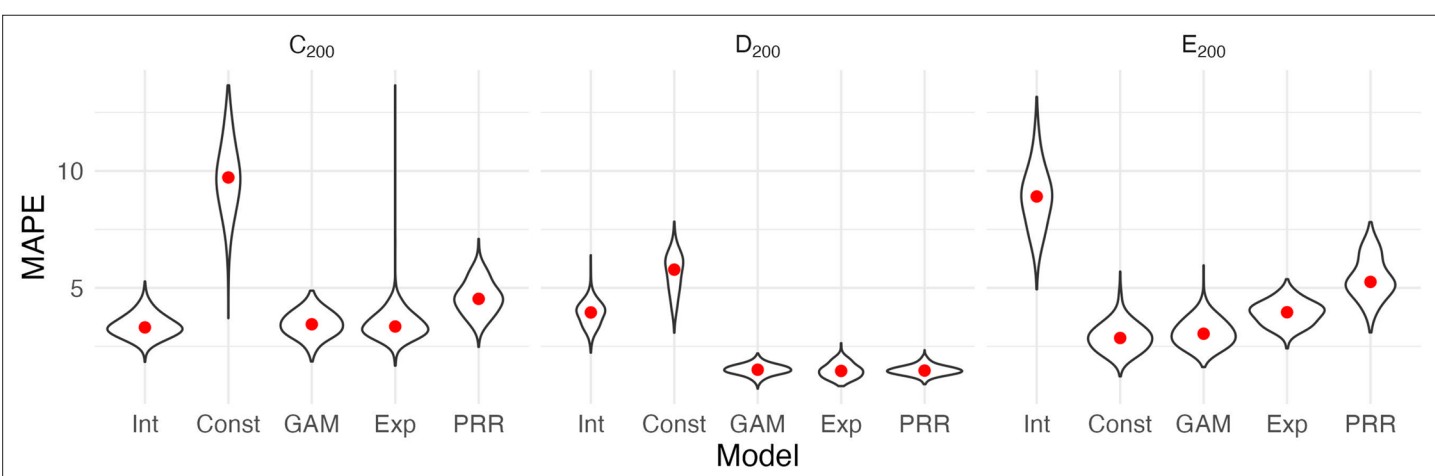

**Figure 3.** Each panel shows the distribution of median absolute prediction errors (MAPE) obtained using cross validation for each of five models. Panels correspond to the generated datasets shown in **Figure 2**. Models compared are an intercept-only model (Int), a generalized additive model (GAM), and the proportional recovery rule (PRR).

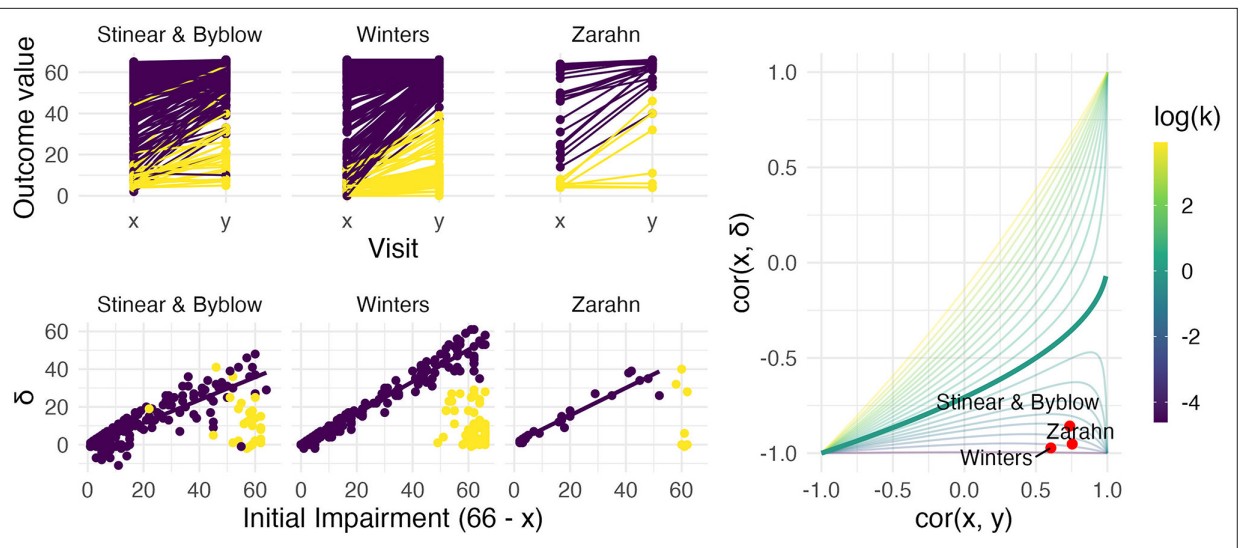

**Figure 4.** The left panels show three real datasets. In the top row, panels show outcome values and baseline ($x$) and follow-up ($y$); points are colored to indicate recoverers (purple) and non-recoverers (yellow) using the definitions from each paper describing the data. In the bottom row, panels show change (delta) against initial impairment ($66 - x$), again separating recoverers and non-recoverers. The right panel shows a contour plot of **Equation 1**, with contours corresponding to values of the variance ratio $k$ and the contour for $k=1$ highlighted. Points on this surface show correlation values obtained for the real datasets.

the (true) constant recovery model performs best. The additive model is flexible enough to capture the underlying non-linear association between $y$ and $x$, and very slightly underperforms compared to constant recovery. The exponential model is a reasonable approximation, but the PRR makes relatively poor predictions.

These results suggest that CV can effectively identify a best (or worst) model for prediction accuracy. When two or more models are similarly accurate, other considerations may be relevant. In Datasets $C_{200}$ and $D_{200}$, for example, the additive model contains the true, simpler model as a special case and is needlessly complex. We also emphasize a limitation of CV, which is the exclusive focus on prediction accuracy. This is analogous to using only $\mathrm{cor}(x, y)$ to understand recovery, rather than $\mathrm{cor}(x, y)$ and $k$ together. As we have discussed elsewhere, dismissing observations like those in Dataset $C_{200}$ as uninteresting because baseline does not predict follow-up would be a mistake: the correlation between baseline and change does not arise from mathematical coupling and may reflect important recovery mechanisms. We therefore suggest CV as one component of a careful analysis.

## Results for reported datasets

We now evaluate three previously reported datasets, described in the Methods section, using the statistical techniques given above.

*Figure 4* illustrates these example datasets. In the left panels, we show observed FM values for patients at baseline and follow-up (top row), and scatterplots of change against baseline (bottom row). Throughout, we differentiate recoverers and non-recoverers as specified in the original analyses. The right panel shows the contour plot of $\mathrm{cor}(x, \delta)$, with points corresponding to observed values among recoverers for each dataset. These values cluster in the bottom right corner, with reasonably large values of $\mathrm{cor}(x, y)$ and $k < 1$.

We next conducted a bootstrap analysis on the subsample of recoverers to obtain inference for $k$ and $\mathrm{cor}(x, y)$. We display the results using 1000 bootstrap samples for each dataset in the contour plot in *Figure 5*, showing the null value corresponding to random recovery for reference, and summarize the results using 95% confidence intervals in *Table 1*. For each dataset, there is strong evidence that $k$ and $\mathrm{cor}(x, y)$ differ from our suggested null values (1 and 0, respectively), and from those corresponding to random recovery. That is, we have evidence both that recovery is related to baseline values in a way that reduces variance at follow-up, and that baseline values are predictive of follow-up. These results inform conclusions about the statistical significance of observed associations between baseline, change, and follow-up values using plausible null hypotheses, and specifically address

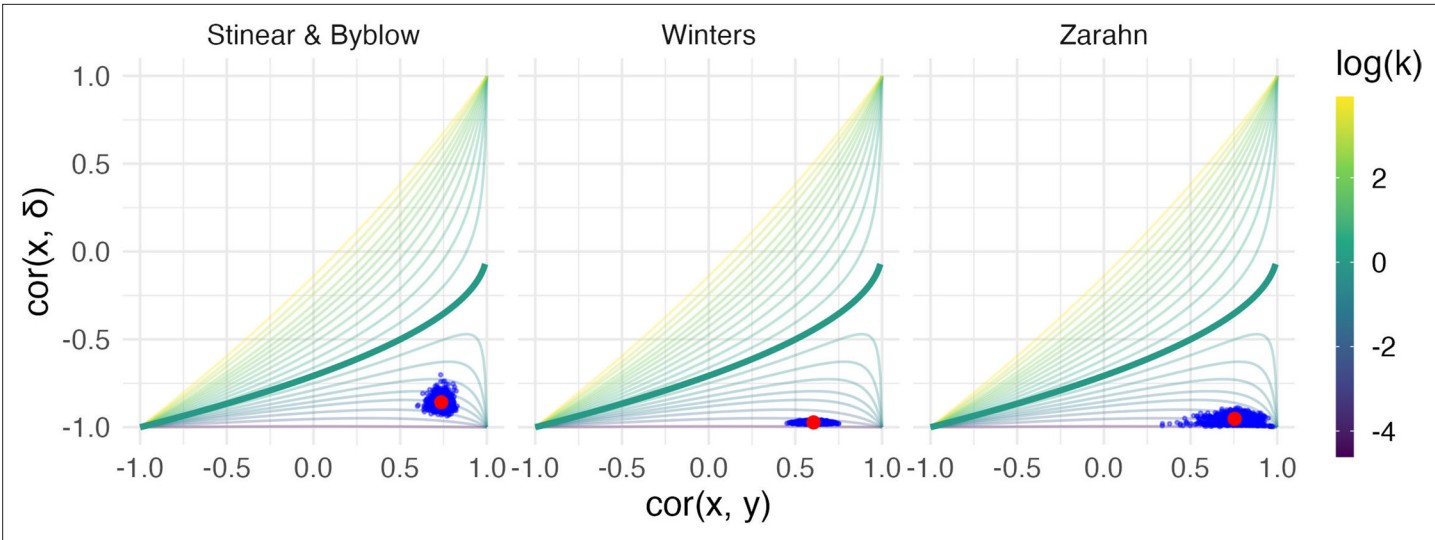

**Figure 5.** Each panel shows the results of the bootstrap procedure used to obtain inferences about the value of correlations and variance ratio. Red points are the values obtained for the full dataset, and blue points are values obtained in each of 1000 bootstrap samples; points are overlaid on the contour plot of *Equation 1*.

concerns about high correlations induced by canonical mathematical coupling. However, several mechanisms (including the PRR and constant recovery in the presence of strong ceiling effects) could give rise to these correlations.

We next compared the performance of five models in terms of predictive accuracy using CV. As for generated data, we considered an intercept-only model, a model assuming constant recovery with a ceiling effect, an additive model, an exponential model, and the PRR. For each dataset, we generated 1000 training/testing splits and summarize prediction accuracy using MAPEs. The plots below show the distribution of MAPE across splits for each model and dataset. In these examples, constant recovery with a ceiling underperforms against competing methods. The exponential model, additive model, and PRR prediction accuracies are often comparable, although the PRR appears superior for the Stinear and Byblow and Zarahn datasets. A figure in Appendix 1 shows fitted values for each model applied to training datasets. In keeping with the results for prediction accuracy, the constant recovery model is visually a poor fit. In some instances, the additive model is too flexible, especially for the Zarahn data. The additive and exponential models are similar to the PRR in terms of fitted values despite their additional flexibility and complexity.

To provide some frame of reference for the MAPEs reported in *Figure 6*, in *Table 2*, we provide values for the percent of outcome variation explained by the PRR for each dataset.

The preceding results suggest that data among recoverers is consistent with the PRR for each study. The strong correlations between baseline and follow-up are not driven by canonical forms of mathematical coupling or regression to the mean. While similar values of $k$ and $\mathrm{cor}\,(x, y)$ can arise from a variety of mechanisms, the PRR clearly outperforms a model that assumes constant recovery in the presence of a ceiling effect in terms of prediction accuracy. Competing models for recovery, specifically those allowing for non-linear association between baseline and follow-up, do not produce more accurate predictions than the PRR, and are similar in their fitted values.

**Table 1.** Values and 95% confidence intervals for $k$ and $\mathrm{cor}(x, y)$ for each of three datasets. Confidence intervals are obtained through a bootstrap procedure with 1000 bootstrap samples.

| Dataset name | $k$ | $\mathrm{cor}\,(x, y)$ |
|---|---|---|
| Stinear and Byblow | 0.39 [0.27, 0.54] | 0.73 [0.66, 0.79] |
| Winters | 0.07 [0.05, 0.09] | 0.59 [0.49, 0.69] |
| Zarahn | 0.13 [0.03, 0.24] | 0.75 [0.55, 0.92] |

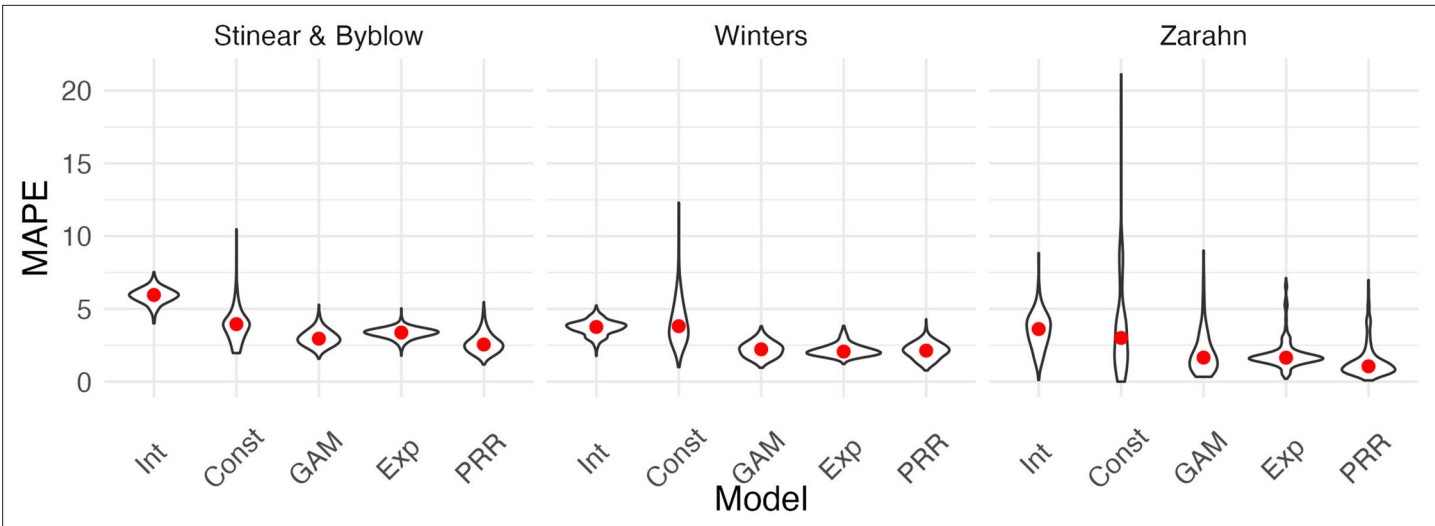

**Figure 6.** Each panel shows the distribution of median absolute prediction errors (MAPEs) obtained using cross validation for each of five models. Panels correspond to the datasets shown in *Figure 4*. Models compared are an intercept-only model, a generalized additive model, and the proportional recovery rule (PRR).

## Distinguishing between recoverers and non-recoverers

To this point, we have focused on recoverers under the implicit assumption that a biologically distinct group of non-recoverers exists and can be identified. This assumption is supported by prior studies of upper limb motor control and deficits in other domains (*Prabhakaran et al., 2008*; *Byblow et al., 2015*; *Winters et al., 2016*; *Zandvliet et al., 2020*), and promising work suggests that it is possible to identify non-recoverers using baseline characteristics (*Byblow et al., 2015*). However, the identification of non-recoverers has not been approached in the same way across studies, and there is concern that data-driven methods can produce misleading results in some circumstances (*Hawe et al., 2018*; *Lohse et al., 2021*). It is necessary to understand the limitations of some methods for data-driven subgroup identification before assessing related arguments about recovery and the PRR.

Clustering refers to a collection of methods for unsupervised learning that has several weaknesses in the context of recovery. It is unclear what clustering approach is best, and results can be sensitive to this choice. Perhaps more critically, determining the 'true' number of clusters present is imprecise and open to interpretation. Although tools like the Gap statistic (*Tibshirani et al., 2002*; *James et al., 2013*) can provide guidance, the choice between one, two, or more clusters is not supported by hypothesis tests, confidence intervals, or other methods for statistical inference. Practitioners are encouraged to try several numbers of clusters and cautioned to be aware that there is rarely a single best selection (*James et al., 2013*). Results are typically considered exploratory and assessed for validity based on visual inspection or additional supporting information. As such, clustering can provide only limited evidence for or against the existence of recoverers and non-recoverers (or finer partitions of patients) when examining a specific dataset. At worst, one might simply assume that distinct recoverer and

**Table 2.** The proportion of follow-up ($y$) variation explained by the proportional recovery rule (PRR) for each of three datasets.

Follow-up fitted $y^{PRR}$ values are obtained by adding predicted change from the PRR to observed baseline values, and we compute $\left( 1 - \frac{\sum (y_i - \hat{y}_i^{PRR})^2}{\sum (y_i - \bar{y})^2} \right)$.

| Dataset name | Percent of outcome variation explained |
| --- | --- |
| Stinear and Byblow | 0.55 |
| Winters | 0.35 |
| Zarahn | 0.56 |

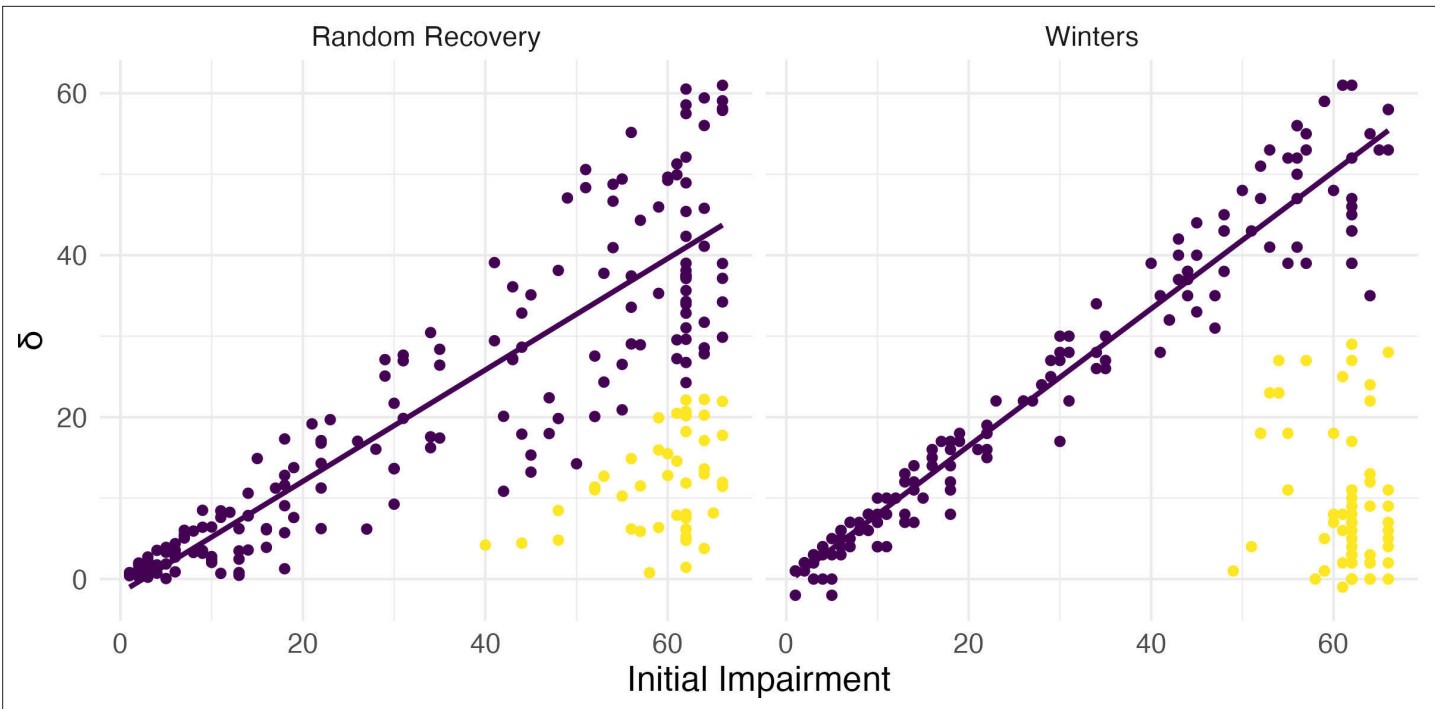

**Figure 7.** Both panels show change between baseline and follow-up against initial impairment (66 – baseline). Left panel shows data generated under a 'random recovery' process, in which outcome values are drawn from a uniform distribution over the baseline value and ceiling. Right panel presents again data from **Winters et al., 2015**.

non-recoverer clusters exist and can be separated using clustering. Failure to critically examine this assumption using visual and quantitative data checks can yield misleading results.

The weaknesses inherent to cluster identification can lead to flawed scientific conclusions. **Hawe et al., 2018**, and **Lohse et al., 2021**, show how errors can arise using 'random' recovery. Put briefly, random recovery assumes that follow-up values $y$ are uniformly distributed between the baseline $x$ and the maximum possible value. If data arise via this mechanism and it is assumed that two groups exist, the use of clustering will often uncover one cluster that appears to follow the PRR with a recovery proportion of roughly 0.75. Building on this observation, **Lohse et al., 2021**, propose an approach for comparing the PRR to random recovery. Data under random recovery can be generated by drawing baseline values $x$ from the observed values with replacement (to mimic the distribution of baseline values in the sample) and then drawing follow-up values $y$ from a uniform distribution between the baseline $x$ and the maximum possible value. This generated data can be analyzed through clustering followed by regression to obtain an observed slope in the 'recoverer' cluster. By repeating this process many times, one can obtain the distribution of slopes in the 'recoverer' cluster under random recovery. Treating this as a null distribution, **Lohse et al., 2021**, argue, provides a way to quantify whether a slope obtained through the same analysis of real data is consistent with random recovery. The authors assert that 'current data do not support the claim that recovery is proportional any more than that recovery is random'.

However, this narrow focus on the distribution of slopes that arises from random recovery bypasses a critical question – whether the results of the clustering analysis themselves are consistent with random recovery. Put differently, one must first ask whether there is evidence for the existence of distinct clusters in observed data using random recovery to generate a null distribution. Viewing data generated under random recovery alongside data obtained in a study of upper limb motor control recovery, as in **Figure 7**, emphasizes this difference. While clustering can misleadingly identify a 'recoverer' group when data actually follow random recovery, visual inspection suggests that an obviously different mechanism underlies the Winters dataset. The null distribution of within-cluster dispersion, rather than slopes, provides a way to quantify this difference; see Methods for details.

Given the contrast between panels in *Figure 7*, it is not surprising that this approach rejects the null of random recovery for these data (p<0.001).

From this, we conclude that an inappropriate application of unsupervised learning (i.e. one that presumes two clusters exists) can produce misleading results. This problem is exacerbated by the lack of concrete statistical guidance for determining the number of clusters in a sample. (Similar issues can arise in other ways, for example from a definition of non-recoverers as those who recover less than expected under the PRR.) Simulations of random recovery, like those in *Hawe et al., 2018*, and *Lohse et al., 2021*, are useful for illustrating these issues but should not be misinterpreted: the ability to produce slopes like the PRR through analyses of generated data does not refute the PRR (or any other model of recovery). Nor does it argue against the existence of meaningful recoverer and non-recoverer groups. In many studies of upper limb motor control recovery, a distinct group of non-recoverers is clear from a visualization of the data (and supported through appropriate hypothesis tests) or can be identified through a distinct biomarker, and does not artifactually arise through clustering.

Finally, we distinguish the identification of a biologically distinct group comprising non-recoverers from the stratification of patients into severe and non-severe groups based on initial impairment. *Zarahn et al., 2011*, for example, used a subjective approach to classify patients with baseline FM ≤ 10 as severely affected. More recently, *Bonkhoff et al., 2022*, used Bayesian hierarchical models and formal model selection approaches to identify a threshold (also found to be FM = 10) above and below which patients exhibit different recovery patterns. Our point in this section, meanwhile, is that the severely affected population contains biologically meaningful subgroups – recoverers and non-recoverers – and that these can be identified using appropriate unsupervised learning methods.

## Discussion

The PRR was discovered while searching for a possible regularity in the relationship between initial impairment, as measured by the FM-UE, and recovery in the context of upper limb paresis in the time shortly after stroke (*Krakauer and Marshall, 2015*). It was understood as a description of the biological change process that underlies observed recovery, and subsequently evaluated in other recovery settings. Although it was not intended to form the basis for patient-level predictions, the strong correlations between initial impairment and recovery has suggested that accurate predictions are possible. At its core, the PRR models the association between baseline and change; this is always a fraught statistical problem, and recent publications on the PRR have revived longstanding concerns in the context of stroke recovery (*Hope et al., 2018*; *Hawe et al., 2018*; *Bonkhoff et al., 2020*).

We have revisited the arguments for the PRR as a descriptive and predictive model, focusing on key statistical questions at each step. We identified scenarios in which observed correlations are 'artifactual' – induced either by mathematical coupling or by regression to the mean due to measurement error – versus those when they are real signals, emphasizing the variance ratio and tests similar to Oldham's method. For non-artifactual signals, we used CV to compare models for recovery (e.g. PRR versus constant recovery with a ceiling); this also provides a concrete metric for the clinical usefulness of predictions made by each method. Finally, we considered the problem of distinguishing recoverers from non-recoverers, and the limitations of unsupervised learning for this problem.

Our findings in these datasets suggest that the association between initial impairment and change is non-artifactual, and the PRR is better as a biological and predictive model than several competing models, including constant recovery with a ceiling effect. These data also suggest that a biologically distinct group of non-recoverers exists. We distinguish between the usefulness of the PRR as a biological and a predictive model deliberately: biological models are important for understanding the recovery process itself, and accurate predictions are important in clinical care. Although the PRR is useful for both, it isn't necessary that a single model address both considerations simultaneously.

We acknowledge several limitations and caveats. The statistical considerations for recovery are nuanced and often counterintuitive. Our arguments deviate from usual null-value hypothesis testing, and recognize that zero is often not the appropriate null value. This is related to the important observation that 'large' correlation values can arise in a variety of settings, which complicates but does not invalidate statistical approaches. While we distinguish between the usefulness of a model as either biological or predictive, our preferred method for comparing models is based only on their predictive accuracy. CV can identify models that have better or worse predictive performance, but in itself does not examine the validity of underlying model assumptions or ensure that better-performing models

are accurate enough to make meaningful clinical predictions. Recoverers and non-recoverers should be identified in a way that is not artificial, and that does not serve to introduce evidence for a model of recovery that would not otherwise exist. Lastly, we recognize that not every dataset will be similar to those we presented; in those cases, we hope the tools we suggest will be used to evaluate the PRR against other models.

Indeed, a recent large-scale cohort study examining the PRR presents data that differ in notable ways from the studies we consider (*Lee et al., 2021*). The authors are concerned about ceiling effects in the FMA-UE, but use an analysis approach that differs from the framework presented here. *Lee et al., 2021*, use a logistic regression to model the probability of achieving full recovery; they observe that more than half of all patients, and more than 60% of patients with baseline FMA-UE above 46, recover to the maximum possible value. These data mimic those in our generated Dataset $E_{200}$, and might benefit from the comparison between the PRR and a non-linear model in terms of predictive accuracy. Certainly, the FMA-UE lacks resolution, especially at higher values, and a refined outcome measure that ameliorates such ceiling effects is needed. There is nothing in this study that precludes the possibility that PRR would apply for this new measure. Meanwhile, we're skeptical of broader claims made regarding the non-generalizability of the PRR. The cohort in *Lee et al., 2021*, is much more mildly impaired at baseline than data we have observed; this could have several causes, but the choice to exclude patients with FMA-UE hand subscore of 0 at day 7 likely removes many moderate and severely affected patients who are unlikely to recover to ceiling. Thus, in a sense the study selected for a cohort that is not only clinically non-representative, but one that makes it hard to derive the PRR by looking for changes at the high-end of the FMA-UE, a range known to fail to detect subtle residual deficits.

The PRR provides an appealingly simple model for understanding recovery, even if the statistical approaches for evaluating it are not direct. We argue that the PRR is and will continue to be relevant as a model for recovery, but we agree with others that more complex analytic strategies are necessary to move beyond it. Recovery depends on factors other than initial impairment. The PRR estimates an average recovery proportion, but individuals will recover more or less than that average; determining whether this is 'noise' or heterogeneity that can be predicted using other forms of measure at baseline is an important next step. Similarly, reliable techniques to identify non-recoverers at baseline are needed. Different outcome measures and different recovery settings may not be well described by the PRR. A specific focus on predicting long-term outcomes is an important goal; supervised learning methods may be helpful in this, although many of these methods have unclear interpretations.

More complex models than the PRR do not inherently resolve the statistical issues we've raised and may in fact make them harder to identify. For example, it's been suggested that mixed models could account for differences in baseline values and change across subjects in a way that avoids mathematical coupling by modeling baseline and follow-up values directly (*van der Vliet et al., 2020*). This is not the case. Consider a model for outcome values at baseline (time = 0) and a single follow-up (time = 1) that includes both random intercepts and random slopes. When such a model is applied to Datasets A, C, and D, the random intercepts and slopes will be correlated; low random intercepts will suggest steeper random slopes in a way that mimics baseline and change scores. Put differently, to predict a follow-up value from baseline, one must use the baseline value to predict a random slope using the correlation between random effects. Indeed, for Dataset A that correlation will equal –0.71 – the same as $\text{cor}\,(x, \delta)$. Distinguishing between mathematical coupling and recovery that changes the variance ratio remains a problem, but now one that involves the correlation of random effects. Alternative coding for the time variable can induce a model that mimics Oldham's method, and avoids coupling based on similar arguments (*Blance et al., 2005*).

Our point is not that mixed models are a wrong choice – in fact, we are enthusiastic about work that combines serial measurement, non-linear trends, random effects, and mixture models (*van der Vliet et al., 2020*). Taken together, these elements can overcome many of the limitations introduced by studies that include only baseline and single follow-up observations. Serial measurements and careful modeling provide insight into patients' recovery trajectory, reduce effects of measurement error, and identify non-recoverers early in the recovery process. Nonetheless, non-linear mixed models fit to serial measurements are not a panacea, and do not avoid the challenges described in this paper. The same fundamental issues that arise from within-subject correlation and changing variances over time due to recovery and measurement ceilings should be considered when using this or any other model

framework. Indeed, more complex models may serve to mask these basic issues by making them implicit rather than explicit.

Allowing for differences in datasets, implementations, and performance criteria, our results are consistent with those reported in recent papers examining the PRR as a predictive model. *Bonkhoff et al., 2020*, fit several models to a subset of patients selected to mitigate ceiling effects and avoid including non-recoverers, and found that that PRR was among the best performing models in terms of out-of-sample prediction accuracy. The percent of outcome variation explained was lower in their work than in our analyses (21% versus 35–56%), leading the authors to suggest that the PRR may be better than other models but insufficient for making clinical predictions in the context of precision neurology. Our view is a less pessimistic take on the same information: that the PRR outperforms competing models (including those intended to account for ceiling effects) attests to its value, and the percent of outcome variation explained is suggestive of an important biological mechanism that should be investigated and understood — even if $R^2$ values are not as high as has been suggested. After all, if it were found that a risk factor accounted for greater than 20% of the variance in the chance of getting a disease it would immediately be investigated and prevention attempted. From the standpoint of clinical prediction, it seems likely that accurate models developed in the future will include initial impairment as a covariate, and may include a term that reflects proportional recovery.

There is growing consensus around the need for careful comparisons of different models for recovery and for analytic strategies that minimize the impact of ceiling effects. *Bonkhoff et al., 2020*, and *Bonkhoff et al., 2022*, use a statistically rigorous model selection approach based on leave-one-out cross-validated deviances, which is particularly well suited to comparing Bayesian hierarchical models. These papers also consider only patients with initial impairments below 45 to avoid ceiling effects induced by mildly affected stroke patients. We use cross-validated MAPE because it explicitly focuses on prediction accuracy, and the results can therefore be interpreted in the context of single-subject predictions. We also hesitate to discard a substantial fraction of our data and miss the opportunity to model recovery for mildly affected patients at baseline, although we recognize that the narrower outcome distribution for patients at or near ceiling can affect measures of model performance. These analytic preferences reflect different approaches to questions and challenges whose importance is increasingly agreed upon.

In light of this, we stress again that the PRR is intended as a model for recovery from a specific form of impairment as measured by the FM, and is best understood as an attempt to mathematically capture a component of the recovery process. The emphasis on change between baseline and follow-up is deliberate: biological recovery is a process that causes a change, which then leads to the final value. Understanding whether recovery varies across patients and what mechanism might drive such variability is a fundamental scientific question. That the PRR also has some value for predicting final outcomes is to be welcomed but not necessary for its biological importance.

Applications of the PRR in studies of upper limb motor control recovery have often found that, averaging across recoverers, roughly 70% of lost function is regained in the time shortly after stroke. We've argued that this is attributable to spontaneous biological recovery (*Cramer, 2008*; *Zeiler and Krakauer, 2013*; *Cassidy and Cramer, 2017*). The existence of such a mechanism does not imply that behavioral interventions are unable to improve patient outcomes. Instead, future clinical trials should seek to improve on the proportion of impairment reduction, reduce the proportion of non-recoverers, or induce changes that are distinct from (and better than) those expected under the PRR.

## Conclusions

Consideration of associations between baseline, follow-up, and change continues to be of interest across scientific domains despite the statistical challenges, and for good reason: these can be and often are related in ways that are not due to artifacts. Our goal here was to clarify statistical reasoning concerning the problem of relating baseline to change in the context of stroke recovery. It is for the reader to decide whether the work presented here is either a bumpy statistical detour or an interesting scenic route. In either case, rigor about this issue is essential for continued biological and clinical progress, and it would be unfortunate if the recent spate of papers leads to dismissal of the PRR out of either confusion or exhaustion.

As a model that relates baseline values to change, the PRR requires the application of careful, and sometimes counterintuitive, statistical techniques. We have described well-established but

non-standard approaches to distinguish between artifacts due to coupling from signals that arise from true associations, introduced tools to compare competing models for recovery based on their predictive accuracy, and elaborated on issues that can arise when using some data-driven methods to distinguish recoverers and non-recoverers. In our analyses of real data, we found evidence for signals not attributable to coupling, and obtained better predictions using the PRR than a model that assumes constant recovery (with some noise) up to a ceiling. This suggests that the PRR is non-artifactual and remains relevant, at the very least, as a model for recovery. Future work should seek to both explain the mechanistic basis for PRR and improve upon it as a predictive model.

## Methods

### Reported datasets

Details of inclusion and exclusion criteria are available in the referenced literature. All patients received usual care according to evidence-based stroke guidelines for physical therapists, but no systematic interventions.

- Stinear and Byblow: combined data from two studies. First, *Byblow et al., 2015*, assessed an a priori prediction of proportional recovery based on corticospinal tract integrity using transcranial magnetic stimulation to determined motor evoked potential status (MEP+, MEP-) for 93 patients within 1 week of first-ever ischemic stroke stroke. FMA-UE were obtained at 2, 6, 12, and 26 weeks. Second, *Stinear et al., 2017*, added data from recurrent ischemic stroke and intracerebral hemorrhage patients with new upper limb weakness to form a larger dataset of 157 patients, all with known MEP status. Following this work, we define recoverers and non-recoverers as MEP+ and MEP-, respectively.
- Winters: first-ever ischemic stroke patients were recruited for the prospective cohort study entitled Early Prediction of functional Outcome after Stroke (EPOS) (*Nijland et al., 2010*; *Veerbeek et al., 2011*). Data comprise baseline (day 2 post-stroke) and follow-up (day 187 post-stroke) FMA-UE observations for 223 patients; 211 were originally reported in *Winters et al., 2015*, with recoverers and non-recoverers identified using a hierarchical clustering based on average pairwise Mahalanobis distances.
- Zarahn: data consist of patients with first time ischemic stroke with some degree of clinical hemiparesis (NIH stroke scale for the arm $\geq$ 1). FMA-UE was assessed both at ~2 days post-stroke and ~3 months post-stroke and were originally reported in *Zarahn et al., 2011*. We focus on the 30 patients in the imaged subsample with publicly available FMA-UE values. Recoverers and non-recoverers are determined using baseline FMA-UE >10.

Ethical approvals were obtained for each dataset; see referenced literature for details. The Winters and Zarahn datasets are included in supplementary materials. The Stinear and Byblow data were collected under ethical approvals that do not permit placing data online. Data can be made available under reasonable request to the author.

### Resampling approach for inference on $\mathrm{cor}\,(x, y)$ and $k$

We suggest a bootstrap procedure to obtain confidence intervals for $\mathrm{cor}\,(x, y)$ and $k$. A single bootstrap sample can be constructed by selecting subjects (pairs of both $x$ and $y$ values) from a full dataset with replacement. Next, we compute the values $k$ and $\mathrm{cor}\,(x, y)$ for the bootstrap sample. This is repeated a large number of times (e.g. 1000) to produce an empirical distribution for the quantities of interest, which can be used to derive corresponding confidence intervals. Simulations evaluating this approach suggest that coverage of 95% CIs is roughly 0.95 for moderate sample sizes.

Our suggested null values for $k$ and $\mathrm{cor}\,(x, y)$ are 1 and 0, respectively, but other values can be used. For null hypotheses framed in terms of data generating mechanisms rather than parameter values, such as the 'random' recovery null hypothesis (*Lohse et al., 2021*), it can be difficult to derive the corresponding null values. In these cases, we suggest to obtain empirical values by generating a large dataset under the null and computing the $k$ and $\mathrm{cor}\,(x, y)$ directly.

### CV to compare models

Data are divided into training and testing sets; training data are used to fit models, and these models are applied to data in the testing set to obtain predicted outcomes. The difference between predicted and observed outcomes in the testing data reflects each model's ability to make accurate predictions

of data not used in model development. Comparing models in terms of their MAPE is an established technique for choosing among candidate models, and the MAPE provides a measure of the anticipated prediction error for a given model.

CV has several possible implementations; here, random training and testing splits are comprised of 80% and 20% of the data, respectively, and the training data is used to fit each model. Given model fits, predictions are made for the testing dataset and compared to actual outcomes, and the difference is summarized using the MAPE. This process is then repeated 1000 times, so that the distribution of MAPE across training and testing splits is obtained.

## Defining a null distribution for within-cluster dispersion

Although the Gap statistic does not allow for inference to determine the number of clusters within a sample, it provides a useful metric for comparing observed data to a null hypothesis of random recovery. Intuitively, for a given number of clusters $k$, the Gap statistic $\mathrm{Gap}(k)$ compares the observed degree of within-cluster similarity to the expected degree of within-cluster similarity under a reference distribution. Suppose a clustering analysis has produced clusters $C_1$, $C_2$, $\ldots$, $C_k$. Assuming the Euclidean distance is used to measure distance between observations in the same cluster, we measure overall within-cluster dispersion using the pooled within cluster sum of squares around the cluster mean:

$$W_k = \sum_{r=1}^{k} \sum_{z_i \in C_r} \|z_i - \mu_r\|^2 .$$

Here, $\mu_r$ is the within-cluster mean and $z_i$ is the vector of observed values for subject $1 \leq i \leq n$ (e.g. $z_i = (x_i, y_i)$).

The value of $W_k$ decreases as $k$ increases, regardless of whether additional true clusters are identified. The Gap statistic therefore standardizes $\log(W_k)$ to provide guidance on the choice of $k$ within a dataset. To standardize $\log(W_k)$, one compares the observed value to its expectation under a reference distribution:

$$\mathrm{Gap_n}(k) = E_n^* \{\log(W_k)\} - \log(W_k) .$$

The expectation $E_n^* \{\log(W_k)\}$ is approximated through the analysis of multiple datasets generated under the reference distribution. For each dataset one computes $\log(W_k^*)$, and the expected value is taken as the average across these. The reference distribution in the calculation of the Gap statistic is often uniform over the range of each feature in the clustering analysis. In the context of recovery, this suggests a square over all possible values of $x$ and $y$ rather than the triangular region defined by random recovery, but the spirit is similar.

We use the Gap statistic to measure the strength within-cluster dispersion against that expected under a reference distribution. To compare observed data to a null distribution, we suggest a resampling-based approach to construct a null distribution for $\mathrm{Gap_n}(k)$. For random recovery, to construct a single resampled dataset, we suggest to sample observed $x$ values with replacement; generate corresponding $y$ values under random recovery; and obtain $\mathrm{Gap_n^*}(k)$. This process is repeated many times, and the observed value $\mathrm{Gap_n}(k)$ is compared to the distribution of $\mathrm{Gap_n^*}(k)$. This process is very similar to one suggested by *Lohse et al., 2021*, with the exception that we focus on evidence for clustering through $\mathrm{Gap_n}(k)$ rather that the distribution of slope values obtained in analyses of resulting clusters.

The Gap statistic can be computed for any clustering method. By extension, our method for comparing the observed amount of within-cluster dispersion to that expected under a null distribution can as well. In our analyses we follow recent literature, and use hierarchical clustering based on Mahalanobis distances between points. Simulations suggest that our testing approach achieves nominal size: when data are in fact generated under random recovery, we reject the null hypothesis 5% of the time under $\alpha = 0.05$.

## Implementation and reproducibility

All analyses were implemented in R; supplementary materials contain source code and RMarkdown documents to reproduce all figures and analyses.

## Acknowledgements

We acknowledge the EXPLICIT-stroke consortium for collecting data. This data collection was supported by funding from the Royal Dutch Society of Physical Therapy and the Netherlands Organization for Health Research and Development (ZonMw; Grant No. 89000001). JG's work was supported in part by NIH-funded R01NS097423.

## Additional information

### Funding

| Funder | Grant reference number | Author |
| --- | --- | --- |
| National Institute of Neurological Disorders and Stroke | R01 NS097423 | Jeff Goldsmith |

The funders had no role in study design, data collection and interpretation, or the decision to submit the work for publication.

### Author contributions

Jeff Goldsmith, Conceptualization, Data curation, Software, Formal analysis, Visualization, Methodology, Writing - original draft; Tomoko Kitago, Andreas Luft, Conceptualization, Writing - review and editing; Angel Garcia de la Garza, Visualization, Writing - review and editing; Robinson Kundert, Conceptualization, Formal analysis, Writing - review and editing; Cathy Stinear, Winston D Byblow, Gert Kwakkel, Conceptualization, Data curation, Writing - review and editing; John W Krakauer, Conceptualization, Data curation, Investigation, Writing - original draft, Writing - review and editing

### Author ORCIDs

Jeff Goldsmith http://orcid.org/0000-0002-6150-8997
John W Krakauer http://orcid.org/0000-0002-4316-1846

### Ethics

Our work reanalyzes several previously reported datasets. These were collected under protocols that ensured informed consent and consent to publish was obtained; details of the protocols and approvals are available in papers originally reporting these datasets.

### Decision letter and Author response

Decision letter https://doi.org/10.7554/eLife.80458.sa1
Author response https://doi.org/10.7554/eLife.80458.sa2

## Additional files

### Supplementary files

• MDAR checklist

• Source data 1. The zip file contains two data sets ("Winters"and "Zarahn"), as well as code that can be used to reproduce all analyses of these datasets.

### Data availability

In this work, we reanalyze previously reported datasets. Two of these (referred to as "Winters" and "Zarahn") are included with the submission as a data file. A third (referred to as "Stinear & Byblow") contain data that were collected under ethical approvals that do not permit placing data online. Data can be made available under reasonable request to the author. Researchers interested specifically in the data referred to as "Stinear & Byblow" should contact Cathy Stinear <c.stinear@auckland.ac.nz> and / or Winston Byblow <w.byblow@auckland.ac.nz>, who serve as custodians of the data on behalf of study participants. Please include a description of the planned analyses and a justification for the use of these data specifically. Commercial research using these data is not permitted. All code used in the analyses of all datasets is provided as part of the submission,

and all data that can be made publicly available (i.e. "Winters" and "Zarahn") are included with the submission.

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

## Appendix 1

### Table of null values

In the table below, we provide values of $\text{cor}(x, \delta)$ for a range of input values $\text{cor}(x, y)$, keeping $k$ fixed at 1. These are a result of *Equation 1* in the manuscript, and are intended to provide context for correlations between baseline and change that arise through coupling rather than as a result of a recovery process that affects the variance ratio.

| $k$ | $\text{cor}(x, y)$ | $\text{cor}(x, \delta)$ |
|---|---|---|
| 1 | 0 | –0.707 |
| 1 | 0.1 | –0.671 |
| 1 | 0.2 | –0.632 |
| 1 | 0.3 | –0.592 |
| 1 | 0.4 | –0.548 |
| 1 | 0.5 | –0.5 |
| 1 | 0.6 | –0.447 |
| 1 | 0.7 | –0.387 |
| 1 | 0.8 | –0.316 |
| 1 | 0.9 | –0.224 |

### Correlations, variance ratios, and regression

A simple linear regression of $\delta$ on $\text{ii} = \text{max} - x$, where $\text{max}$ is the maximum possible value of the scale and $\text{max} - x$ is initial impairment, can be written

$$\delta = \beta_0 + \beta_1 \text{ii} + \epsilon.$$

The following expressions relate regression parameters and diagnostics to $\text{cor}(x, \delta)$, $\text{cor}(x, y)$ and $k$:

- $\widehat{\beta}_1 = 1 - \text{cor}(x, y)\sqrt{k}$
- $R^2 = \text{cor}(x, \delta)^2 = \left(\dfrac{\widehat{\beta}_1}{\sqrt{k - 2\widehat{\beta}_1 - 1}}\right)^2$

The next subsection contains derivations of these expressions. From these, we conclude first that $\text{cor}(x, y) = 0$ implies $\widehat{\beta}_1 = 1$. This suggests that usual hypothesis tests of the slope which assume a null value of 0 should instead assume a null value of 1. Moreover, this test measures the association between baseline and follow-up, and slopes that differ significantly from one suggest that baselines can be used to predict follow-up values. Second, we see that the amount of variation explained depends on both $\text{cor}(x, y)$, through the estimated slope, and on the variance ratio; for example, in the canonical example of mathematical coupling, $R^2 = 0.5$. Resampling-based tests for the slope and $R^2$ can be performed analogously to those for $k$ and $\text{cor}(x, y)$ using these (or other) null values.

Like $\text{cor}(x, \delta)$, $R^2$ should be interpreted with caution: the preceding expression provides a way to determine the expected or null $R^2$ for a given slope and $k = 1$, which can be used as a frame of reference for observed $R^2$ values. $R^2$ values that depart from the null value may suggest $k \neq 1$ and, by extension, changes that are related to baseline in a way that systematically reduces the variance ratio. As in the previous section, based on this analysis it will remain unclear whether a statistically significant difference is attributable to proportional recovery, ceiling effects, or some other process; distinguishing between models is the subject of later sections.

The above expression for $R^2$ is in the context of a regression of change on initial impairment, which will differ from the $R^2$ arising from a regression of the follow-up value on the baseline observation. Because these $R^2$ values are the square of $\text{cor}(x, \delta)$ and of $\text{cor}(x, y)$, respectively, much of our previous discussion applies here. The initial value $x$ may explain a higher proportion of variation in $\delta$ than in $y$; this setting is not necessarily artifactual, and high $R^2$ in the regression of change on baseline can be

important even when $R^2$ in the regression of follow-up on baseline is low, just as high cor $(x, \delta)$ can be important even when cor $(x, y)$ is low.

The inclusion of the intercept $\beta_0$ in the simple linear regression differs from the usual formulation of the PRR and helps to establish clear connections between a correlation-based and a regression-based perspective. This is helpful for evaluating the results of past studies, especially results expressed in terms of percent variation explained, and refines the use of both correlations and regressions as evidence for the PRR.

Similar to correlations and variance ratios, results from a regression-based analysis can, when understood correctly, help assess evidence for the PRR in a given dataset: establishing statistical significance using appropriate tests, along with evaluation of regression diagnostics, will suggest whether data are consistent with the PRR. However, all these approaches measure only linear associations, and neither compare the PRR to alternative models for recovery nor evaluate the ability of the PRR to make accurate predictions about patient outcomes.

## Derivation of expressions for $\widehat{\beta}_1$ and $R^2$

In the preceding section, we discuss a simple linear regression of $\delta$ on $\mathrm{ii} = \max - x$, where max is the maximum possible value of the scale and $\max - x$ is initial impairment. This regression can be written

$$\delta = \beta_0 + \beta_1 \mathrm{ii} + \epsilon.$$

The inclusion of an intercept in the simple linear regression differs from the usual formulation of the PRR, but we find model helpful because it connects correlations to regression parameters and summaries. Specifically, the following expressions relate regression parameters and diagnostics to cor $(x, y)$ and $k$:

- $\widehat{\beta}_1 = 1 - \text{cor}(x, y) \sqrt{k}$

- $R^2 = \text{cor}(x, \delta)^2 = \left( \dfrac{\widehat{\beta}_1}{\sqrt{k - 2\widehat{\beta}_1 - 1}} \right)^2$

In a simple linear regression, the OLS estimate of the intercept is the ratio of the covariance of the predictor and response and the variance of the predictor. In this specific regression, we have:

$$
\begin{aligned}
\widehat{\beta}_1 &= \frac{\text{cov}(\delta, \mathrm{ii})}{\text{var}(\mathrm{ii})} \\
&= \frac{\text{cov}(y - x, \max - x)}{var(\max - x)} \\
&= \frac{\text{cov}(y, -x) + \text{cov}(-x, -x)}{\text{var}(x)} \\
&= \frac{\text{var}(x)}{\text{var}(x)} - \frac{cov(y, x)}{\text{var}(x)} \\
&= 1 - \frac{\text{cor}(x, y) \sigma_x \sigma_y}{\text{var}(x)} \\
&= 1 - \text{cor}(y, x) \frac{\sigma_y}{\sigma_x} \\
&= 1 - \text{cor}(y, x) \sqrt{k}
\end{aligned}
$$

Next, we note that in a simple linear regression, $R^2$ is the squared correlation between the outcome and response. Then:

$$
\begin{aligned}
R^2 &= \text{cor}(ii, \delta)^2 \\
&= \text{cor}(\max - x, \delta)^2 \\
&= \text{cor}(x, \delta)^2
\end{aligned}
$$

Lastly, for linear regressions that include an intercept, $R^2$ is the squared correlation between the outcome and fitted values obtained from the model. Starting from this, we find:

$$
\begin{aligned}
R^2 &= \text{cor}\left(\widehat{\beta}_0 + \widehat{\beta}_1 ii, \delta\right)^2 \\
&= \left(\frac{cov\left(\widehat{\beta}_0 + \widehat{\beta}_1 ii, \delta\right)}{\sqrt{var\left(\widehat{\beta}_0 + \widehat{\beta}_1 ii\right) var(\delta)}}\right)^2 \\
&= \left(\frac{cov\left(\widehat{\beta}_0 + \widehat{\beta}_1 (max - x), y - x\right)}{\sqrt{var\left(\widehat{\beta}_0 + \widehat{\beta}_1 (max - x)\right) var(y - x)}}\right)^2 \\
&= \left(\frac{cov\left(-\widehat{\beta}_1 x, y - x\right)}{\sqrt{var\left(-\widehat{\beta}_1 x\right) var(y - x)}}\right)^2 \\
&= \left(\frac{cov\left(-\widehat{\beta}_1 x, y - x\right)}{\sqrt{var\left(-\widehat{\beta}_1 x\right) var(y - x)}}\right)^2 \\
&= \left(\frac{\widehat{\beta}_1 cov(-x, y - x)}{\widehat{\beta}_1 \sqrt{var(x) var(y - x)}}\right)^2 \\
&= \left(\frac{var(x) - cov(x,y)}{\sqrt{var(x)\left[var(x) + var(y) - 2cov(x,y)\right]}}\right)^2 \\
&= \left(\frac{var(x) - cor(x,y)\sqrt{var(x) var(y)}}{\sqrt{var(x)\left[var(x) + var(y) - 2cor(x,y)\sqrt{var(x) var(y)}\right]}}\right)^2 \\
&= \left(\frac{var(x) - cor(x,y) var(x) \sqrt{k}}{\sqrt{var(x)\left[var(x) + var(x)k - 2cor(x,y) var(x)\sqrt{k}\right]}}\right)^2 \\
&= \left(\frac{var(x)\left(1 - cor(x,y)\sqrt{k}\right)}{var(x)\sqrt{1 + k - 2cor(x,y)\sqrt{k}}}\right)^2 \\
&= \left(\frac{1 - cor(x,y)\sqrt{k}}{\sqrt{k + 2 - 2cor(x,y)\sqrt{k} - 1}}\right)^2 \\
&= \left(\frac{\widehat{\beta}_1}{\sqrt{k - 2\widehat{\beta}_1 - 1}}\right)^2
\end{aligned}
$$

## Specification of the GAM

We use a GAM to allow smooth, non-linear associations between baseline $x$ and follow-up $y$. This is intended to provide a flexible candidate model for comparison with other mechanistic model implementations, including the PRR and constant recovery in the presence of strong ceiling effects.

Our implementation uses the `gam` function in the `mgcv` R package (**Wood, 2012**). A thorough overview of the theoretical and practical underpinnings of this widely used package can be found in related monograph (**Wood, 2017**). Briefly, the default gam approach estimates non-linear associations using a rich thin-plate spline expansion with an explicit penalization to enforce smoothness in the result; for intuition, a high degree of penalization results in linear fits, while less penalization allows for greater non-linearity. The smoothing parameter(s) are selected using generalized CV as part of the model-fitting procedure.

Code supplements contain all model fitting procedures; for clarity, we note that our implementations are of the form: mgcv::gam (y ~ s(x), data = winters_df).

## Fitted values across training/testing splits

We use CV to compare the performance of five models in terms of predictive accuracy. We considered an intercept-only model, a model assuming constant recovery with a ceiling effect, an additive model, an exponential model, and the PRR. For each dataset, we generated 1000 training/testing splits; results for prediction accuracy using MAPEs are shown in **Figure 6** in the main paper. The **Appendix 1—figure 1** shows fitted values for each model (except the intercept-only model) applied to training datasets generated in the CV procedure. The constant recovery model is visually a poor fit, and occasionally the additive model is too flexible, especially for the Zarahn data. The

additive and exponential models yield fitted values that are similar to the PRR despite their additional flexibility and complexity.

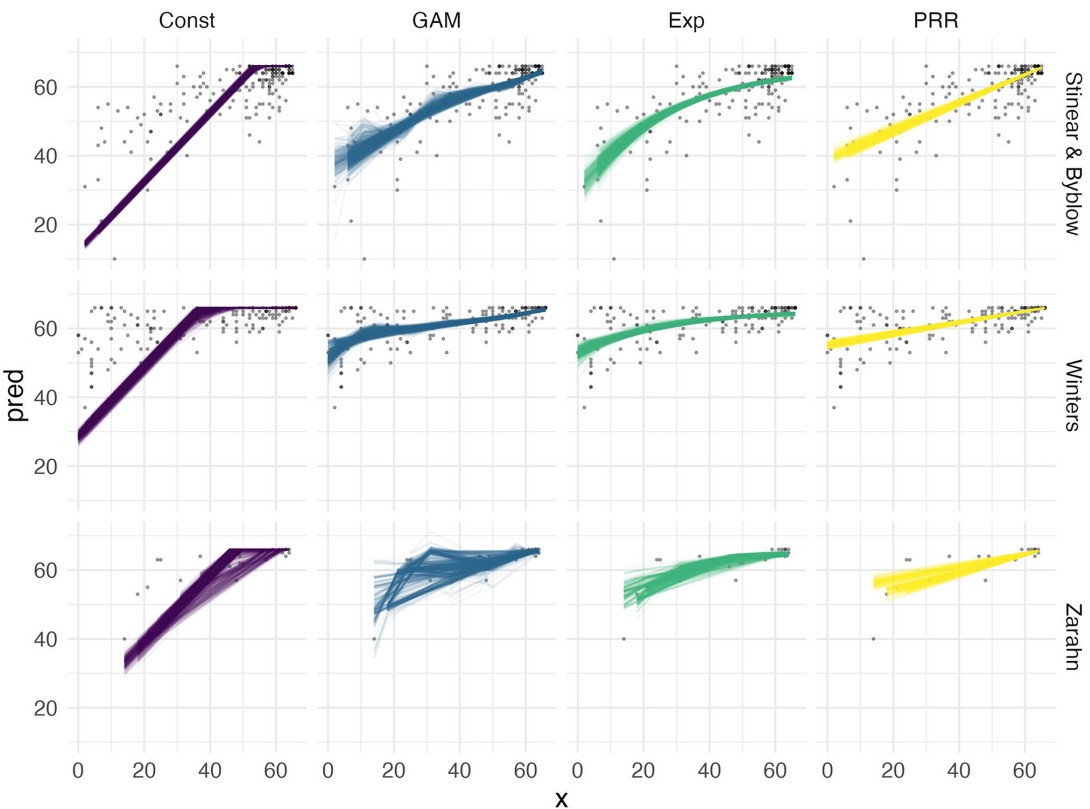

**Appendix 1—figure 1.** Fitted values for each of four models, obtained within separate training / testing splits as part of the cross validation procedure.

