## [Editor Report]

This fundamental work provides a comprehensive look at validity of the Proportional Recovery Rule, which states that patients will recover a fixed proportion of lost function after stroke. By undertaking a thorough investigation of the statistical properties of the analysis of change and baseline values the authors elucidate the statistical framework that can be used regardless of the topic of study. In a compelling model comparison across several large sets of data, the authors confirm support for the Proportional Recovery Rule over other models of recovery.

---

## [Decision Letter]

**Decision letter after peer review:**

[Editors’ note: the authors submitted for reconsideration following the decision after peer review. What follows is the decision letter after the first round of review.]

Thank you for submitting the paper "The proportional recovery rule redux: Arguments for its biological and predictive relevance" for consideration by *eLife*. Your article has been reviewed by 2 peer reviewers, and the evaluation has been overseen by a Reviewing Editor and a Senior Editor. The following individuals involved in review of your submission have agreed to reveal their identity: Howard Bowman (Reviewer #3).

Comments to the Authors:

We are sorry to say that, after consultation with the reviewers, we have decided that this work will not be considered further for publication by *eLife*.

The reviewers had significant concerns about both the clarity of the manuscript and the methods employed. They felt that there remained confounds in the analyses that cannot be sufficiently ruled out. Ultimately, there was not sufficient enthusiasm that the manuscript would have broad appeal to the readership of *eLife*.

*Reviewer #2 (Recommendations for the authors):*

This paper addresses the inherent potential of motor recovery after stroke. The authors would like to validate the concept of proportional recovery which assumes that stroke patient typically show a fixed amount of recovery of about 70% irrespective of where they start. The authors suggest that after controlling for possibly confounding variables like ceiling effects or mathematical coupling, they still could validate the proportional recovery rule. The Results section is difficult to read for a non-statistician. However, the findings reported in the paper are rather incremental than novel. Most importantly, there are no data that explain the neurobiological basis of proportional recovery. Hence, the novelty of this submission remains low apart from more sophisticated statistical models which ultimately come to a similar conclusion than reported years before.

As stated above, the Results section is very difficult to read and appears somewhat lengthy. Lesion analyses or any other data about the possible neurobiological foundations of the proportional recovery are missing. I acknowledge the relatively large sample size, but the Fugl-Meyer score based view without any further para-clinical evidence is somewhat meager. Therefore, I do not see that this paper has a major impact on the field, as in the best case it confirms the proportional recovery rule which has been published many years before.

*Reviewer #3 (Recommendations for the authors):*

This paper presents a defence of the proportional recovery theory of how stroke patients with upper limb motor impairments recover. This is a theory that has come under attack due to confounds associated with the analyses used to assess the theory. The paper acknowledges these confounds, which include compression (to ceiling) enhanced mathematical coupling and over-fitting in the differentiation of recovers and non-recovers. The paper then presents a number of methods: use of non-zero nulls; focus on the variance-ratio; bootstrapping to assess variability in parameter and model fits; comparison of model fits; etc. These certainly improve upon the methods classically used in the literature to justify the proportional recovery hypothesis. Additionally, we appreciate the constructive tone of this paper and its effort to arrive at a consensus position on the proportional recovery issue. Indeed, the overall conclusion that there is a proportional (to lost) recovery pattern amongst the moderately impaired group is a position that we have sympathy with, although that pattern is much weaker than some previous claims in the literature.

However, there seem to be remaining problems with the analyses employed, although, it is difficult to definitively judge the methods employed without more detail than the paper currently offers.

The two most important issues that seem to be weaknesses are as follows.

1) Remaining confounds with key measures: Goldsmith et al. argue that particular combinations of correlation between baseline and follow-up, i.e. cor(x,y); correlation between baseline and change, i.e. cor(x,δ); and the variance-ratio, i.e. k, is diagnostic of proportional recovery (PPR). In particular, they argue that the combination of cor(x,y), cor(x,δ) and k shown in table 1 and figure 5 justify their claim that the three datasets they consider, Stinear and Byblow, Winters and Zarahn, definitively exhibit a PPR pattern. However, the simulations presented in Bonkhoff et al. (2020), show that this same combination of measures can be exhibited by a constant recovery model in the presence of ceiling. Indeed, this is the fundamental confound described by Hope et al., Bonkhoff et al. and Bowman et al., which led to the conclusion that the only way the ceiling confound could be avoided is by throwing data points at ceiling away. (This is actually what I believe you should do with the data in this paper. Note, in Bonkhoff et al. (2020), we verified with model recovery simulations that the throwing away procedure enables the correct model to be recovered.) To make this completely clear, I have prepared a figure, which I am assured will be attached with this review.

https://submit.elifesciences.org/*eLife*_files/2021/09/17/00098237/00/98237_0_attach_15_32737_convrt.pdf

This shows table 1 and figure 5 from Goldsmith et al. on the left and figure 6 from Bonkhoff et al. on the right. I have added annotations to figure 6, which show where the Stinear and Byblow and Winters data sets would sit in the panels for constant recovery. The Zarahn data set does not so obviously correspond to a vertical line, but fits with the general combination of measurements.

2) Limitations of model fits: certainly, the effort to fit models to the three data sets is welcome. However, there is a concern that this model comparison is not a completely fair test of competing theories to proportional recovery. The model fitting comparison in figure 6 is interesting to see. Although, I do find it difficult to interpret the findings, for the following reasons: (a) a criterion has not been given to judge when one model is, in a statistical sense, doing better than another model. I would have expected an assessment of where the MAPE for the best model sits in the bootstrap distribution of MAPEs for each of the other models. This would enable an inference of the kind that the MAPE of the best model is beyond the 95% confidence level of the second best model, and similar for the third. Something of this kind was done in Bonkhoff et al. (2020). (b) I suspect the reason that the Generalised Additive Model does not win is because it is too flexible. It would have been very useful to have seen (perhaps in an appendix) the exact functional form of the Generalised Additive Model. So, I cannot be sure, but I suspect it was not set up to be specifically for saturation at ceiling patterns, with a positive slope leading to a saturation on the right. Consequently, I suspect that the model overfits for many of the bootstrap samples. van der Vliet et al. (2020) give a strong precedent for using a simple exponential, which I suspect would fit the data much better. (c) when I look at figure 4, left panels, lower row, I feel that I do see a strong ceiling pattern in the data. However, this would be much easier to assess if simple x against y (baseline against followup) scatter plots were also presented and model fits for each model were depicted on the scatter plot. This could for example go into an appendix.

van der Vliet, R., Selles, R. W., Andrinopoulou, E. R., Nijland, R., Ribbers, G. M., Frens, M. A., … and Kwakkel, G. (2020). Predicting upper limb motor impairment recovery after stroke: a mixture model. Annals of neurology, 87(3), 383-393.

I have the following more specific comments for the authors.

Abstract: the following statement is made," We describe approaches that can assess associations between baseline and changes from baseline while avoiding artifacts either due to mathematical coupling or regression to the mean due to measurement error". For reasons justified above, and below, I am doubtful that this statement and similar statements made elsewhere in this paper are appropriate.

Introduction section, paragraph beginning "The growing meta-literature discusses …", statement: "the nuanced and sometimes counterintuitive statistical arguments are critical to get right for the sake of furthering our understanding of the biological mechanisms of recovery." I could not agree more than I do with this statement.

Results section, paragraph beginning "A broader argument relates to settings..", with regard to the reference to Bonkhoff et al. (2020) in this paragraph, the reference to "degenerate" is actually most explicitly made in Bowman et al. (2021).

Results section, subsection "Distinguishing true and artifactual signals", paragraph beginning "The value of cor(*x*, δ) depends on the variance ratio *k* and..", statement: "The variance ratio can be used as a measure of the extent to which recovery depends on baseline values, regardless of the value of cor(*x*, *y*)." As indicated above, if this statement is suggesting that the variance-ratio can be used to unambiguously identify a proportional recovery pattern, we do not agree. Indeed, the variance-ratio could be small even when there is no recovery at all; that is, if the mean at follow-up is the same, or indeed below, the mean at baseline.

Results section, subsection "Distinguishing true and artifactual signals", paragraph beginning "All datasets have *x* values generated from a Normal distribution…", how is ceiling handled in these simulations – I would have expected that sometimes a followup score there would be, "by chance", above 66. Can you add a explanation of what happens to these?

Results section, subsection "Distinguishing true and artifactual signals", a minor frustration is that the contour plot on the right of figure 1 is the opposite way around to the surface plot in Hope et al., with a baseline by follow-up correlation of one furthest to the right. At the least, could this discrepancy be mentioned in the caption to help the reader.

Results section, subsection "Distinguishing true and artifactual signals", para beginning "Dataset A is a canonical example of mathematical coupling", I am unsure about quite a lot of the points made in this paragraph. It seems to me that a lot of what is said in this paragraph does not sit well with point 1 above.

Results section, subsection "Distinguishing true and artifactual signals", para beginning "Dataset D represents the least controversial..": I would agree that dataset D is a clear example of proportional recovery. However, our central point in previous publications is that on the basis of the methods typically used, including in this paper, that judgment has to be made informally on the basis of visual inspection. This is because one can obtain exactly the same values of statistical measures (r(X,Y), r(x,δ), k), in a dataset without PRR, but which contains a strong ceiling effect, and indeed, datasets do typically exhibit strong ceiling effects; that is, from what I have seen, empirically collected datasets do not typically look like dataset D, because of the ceiling effect.

Results section, subsection "Distinguishing true and artifactual signals", para beginning "Taken together, the..", text: "We also identify a setting, typified by Dataset D, in which each measure suggests the presence of a relevant association. It is not the case that data like these necessarily imply that recovery follows the PRR. Other biological models could produce data similar to Dataset D, and how to compare competing models will be considered in later sections." Just to say, I was a little confused here. What you are saying here seems to be inconsistent with what you said in the previous paragraph in re. what can be deduced from dataset D.

Results section, subsection "Recasting Oldham's method", paragraph beginning "Instead of cor(*x* + *y*, *x* − *y*), we prefer to focus…": again, some of what is said in this paragraph seems to ignore the confounds that arise from ceiling effects.

Results section, subsection "Correlations, variance ratios, and regression", paragraph beginning "The following expressions relate..", Would it be possible to see a derivation of these two equations? These are not completely standard since \δ = y-x and ii = max-x. This could go in an appendix.

Results section, subsection "Comparing distinct biological models for recovery", paragraph beginning "To illustrate how prediction accuracy…", last sentence: can you give more details of how ceiling is enforced?

Results section, subsection "Comparing distinct biological models for recovery", paragraph beginning "We consider three models for the association …", you say that "Second, we implement the PRR (without intercept) to estimate δ given *x*, with *y* taken to be *x* + δ.", Don't you need to tell us the slope in this eqn, in order to know it is proportional to lost? This is required to rule out proportional to spared or constant recovery. Also, you refer to using a "generalized additive model". Could you give more details of the functional form of this model, perhaps in an appendix? At the least, could you include a relevant reference?

Results section, subsection "Results for reported datasets", paragraph beginning "We next conducted the bootstrap analysis on the subsample…": again, the confound created by ceiling effects impacts this paragraph.

Results section, subsection "Results for reported datasets", paragraph beginning "We compared the performance of three models …": this is where my second point above applies.

Results section, subsection "Results for reported datasets", table 2 caption. Sorry for being dumb, but could you give more explanation of how the R-squared is being calculated here. In particular, could you confirm whether the R-squared equation given in the "Correlations, variance ratios, and regression" subsection is the one being applied here. If it is not this one, can you give the equation that is being used. This is a key issue, since the R-squared values here are much smaller than those given in relevant papers published for some of these data sets.

Results section, subsection "Results for reported datasets", paragraph beginning "The preceding results for …": as previously discussed, the central finding of the work of Hope et al. and Bonkhoff et al. is that, in the presence of ceiling, recovery patterns completely different to the PRR, for example proportional to spared function, can look like PRR, when the measures considered in this paper, cor(\δ, x), cor(x,y), var ratio, are taken. How has the work presented here countered that criticism? I cannot see that it has.

Discussion section, paragraph beginning "Indeed, a recent large-scale cohort study …", it is stated that, "Where Hope et al. (2018) and Bowman et al. (2021) note ceiling effects can compress scores in a way that induces a variance reduction, Lee et al. (2021) observe more than half of all patients, and more than 60% of patients with baseline FMA-UE above 46, recover to the maximum possible value." I do not understand what is being said here. Lee et al. is put in opposition to Hope et al. and Bowman et al., but isn't the Lee et al. ceiling effect going to exactly lead to a reduction in variance at followup?

Discussion section, paragraph beginning "More complex models than the PRR do.." What is being said in this paragraph about Bowman et al. (2020) is not completely clear to me and this may well be a presentational failure on our part when writing Bowman et al. The only place in Bowman et al. (2020) where we refer to mixed models is the following statement, "In particular, van der Vliet et al. present an impressive Bayesian mixture modeling of Fugl-Meyer upper extremity measurements following stroke. Importantly, the authors avoid the confounded correlation of initial scores with change by simply fitting their models to behavioral time-series, that is, raw Fugl-Meyer upper extremity scores, without involving the recovery measure."

However, in the next paragraph, you commend van der Vliet et al. for the same piece of work that we are referring to. Indeed, we were under the impression that the above quoted statement in Bowman et al. (2020), was only a reiteration of what van der Vliet et al. say themselves, which is as follows, "Our current longitudinal mixture model of FM-UE recovery, as opposed to the proportional recovery model, cannot be confounded by mathematical coupling. Hope et al. showed that the correlations between baseline FMUE score (distribution X) and the amount of recovery defined as endpoint FM-UE minus baseline FM-UE (distribution Y-X) found in proportional recovery research could be inflated by mathematical coupling. However, because mathematical coupling applies to correlations of data points (baseline and endpoint FM-UE) and not to models of longitudinal data, the recovery coefficients in our research represent nonconfounded measures of recovery as a proportion of potential recovery. In addition, mathematical coupling does not apply to the outcomes of the cross-validation, as we report correlations between the model predictions and the observed values for endpoint FM-UE and ΔFM-UE rather than correlations of the form X and Y-X."

This leaves us confused, as to what about Bowman et al. (2020) is being criticised in your paragraph beginning "More complex models ….".

[Editors’ note: further revisions were suggested prior to acceptance, as described below.]

Thank you for resubmitting your work entitled "The proportional recovery rule redux: Arguments for its biological and predictive relevance" for further consideration by *eLife*. Your revised article has been evaluated by Michael Frank (Senior Editor) and a Reviewing Editor.

The manuscript has been improved but there are some remaining issues that need to be addressed for further consideration, as outlined below:

Essential revisions:

1) Please expand your discussion of the similarities between the current work and previous critiques of the PRR. This should include a brief comment on the justification of some of the analysis choices (e.g. inclusion/exclusion of patients at ceiling) and the relevance of the findings presented here to other areas of research.

2) Address the statistical concerns raised by Reviewer #2

3) Based on some of the reviewers' comments, you may decide to move some of the analyses to a supplement. This is at your discretion.

*Reviewer #1 (Recommendations for the authors):*

The authors have completed an extensive reworking of their paper, which does a good job of responding to the majority of my comments. Perhaps most importantly, the authors have convincingly countered the argument that I made in my first review that their findings could have arisen from a Constant model with a ceiling. It is excellent to see this rebuttal of my concern. So, I am basically happy with this revision.

It is again good to see the emphasis in this paper on model comparison – that certainly seems to be the right way to go. I also note that the findings in this paper are somewhat consistent with what we have reported in Bonkhoff et al., 2020 – a PRR effect, although considerably weaker than had previously been claimed. So, as the authors point out, there does seem to be convergence in the field, which is excellent.

It remains a pity that the claim that PRR is definitively better on the basis of the presented model fits is still a weak claim. In particular, violin plots for GAM, Exp and PRR are heavily overlapping in figure 6. As you surely know, what is needed is a more sophisticated model comparison, most likely of a Bayesian variety, using something like the model evidence, free energy or deviance information criterion.

The key inference suggested in the paper hinges on an Occams razor argument that if one cannot distinguish between GAM, Exp and PRR, one should pick PRR, since it is more simple. [Although note, I do not believe that Exp is a supermodel of PRR, i.e. I don't think one can get the PRR pattern from Exp, which slightly complicates this line of argument.]

I think that in the absence of a more sophisticated model comparison, of the kind I just highlighted, the Occams razor argument probably is the best one can do. However, since the cross validation is out-of-sample, the flexibility of the model is actually taken into account there.

Obviously, it would be great if this issue could be resolved with a more sophisticated model comparison in a further re-write, however, I don't think this is fair to require at this stage of the review process, but it would be good to acknowledge this weakness in the discussion, and note that competitor approaches are doing better in this respect.

Specific points:

I did not pick this up in the first review, but I spent a lot of time on this review being confused by the right panel of figure 1, which re-represents the surface plot from Hope et al. (2018). I'm now pretty sure your colour bar needs to be inverted. As it is, the yellow contour corresponds to the degenerate region in Hope et al's surface plot, i.e. it does not matter what the value of cor(x,y) is, you always get a correlation of x with change near to -1. However, shouldn't that be for low values of the variability ratio (i.e. log(k) substantially smaller than zero)? Your yellow contour corresponds to log(k) much *bigger* than zero. This issue is also present in Figures 4 and 5.

Top of page 8: "which when baseline (*x*) and follow-up (*y*) are uncorrelated and have the same variance." Should the "when" be deleted here?

Page 9, towards bottom: "relationship between of".

Page 10, 2nd para: "data represent in which baseline values".

Page 17, beginning of last para: you talk about there being five models, but in the next sentence you only list 4.

Page 26: you reference Lohse et al. 2021, but I couldn't find this in the bibliography.

Page 38 top of page: there are a couple of typos here. Also, you again talk about there being five models, but in the next sentence you only list 4.

*Reviewer #2 (Recommendations for the authors):*

1. I think this paper could be made more succinct. It is quite long because the authors used many examples to demonstrate their arguments. Although I feel this approach is well-suited to clinical audience, I am not sure whether the targeted readers may get lost in the technical discussion. For example, I am not sure using bootstrapping is necessary, as we can either directly test the spurious correlation against a more appropriate null hypothesis or using simulation (see the paper in Eur J Oral Sci 2005; 113: 279-288).

2. There are two major issues involving in assessing the relation between change and the baseline. The first is the hypothesis testing and the other is the prediction. Because of mathematical coupling, the usual null hypothesis that the correlation coefficient is zero is inappropriate. This is because the distribution of correlation coefficient is in a restricted space defined by the correlation between x and y, as shown in Figure 1. By the way, I think the paper by Bartko, J. J. and Pettigrew, K. D. (The Teacher's Corner: A Note on the Correlation of Parts with Wholes. Am Stat 22, 41-41, 1968) should be cited, as they are the first to produce such a figure for the correlation between change and the baseline. Another paper of interest is the one by Sutherland, T. M. (The correlation between feed efficiency and rate of gain, a ratio and its denominator. Biometrics 21, 739-749, 1965). Oldham's method is to address this issue of wrong null hypothesis by testing the difference in the variances of X and Y. Other methods are also available as discussed in my previous paper (Tu and Gilthorpe 2007) cited by the authors. As discussed by the authors, Oldham's method or any others based on the same idea has its limitation, if data may be truncated. A possible solution is to design a more sensitive tool to measure the outcome.

3. For the prediction, I think we need to be cautious to use R2 for comparing the performance of different studies. Because R2 is the square of r, i.e. the correlation coefficient. Consequently, the range of R2 is also restricted by the same conditions as r is. Therefore, if different studies have different correlations between X and Y, I do not think their R2 are directly comparable.

4. Moreover, a high R2 does not necessarily mean that patients' recovery can be precisely predicted. Suppose a zero correlation between X and Y, the R2 for using X to predict Y – X will be 0.71^2 = 0.5. Although half of the change's variance can be predicted by the model (i.e. by the baseline value X), this model is useless. This is because R2=0.5 is what we would expect from two sets of random numbers. The baseline and the follow-up values behave like two random variables with zero correlation, this means that X has no use for predicting Y.

5. Regarding to subgroups of patients with different responses, the authors are correct in giving warnings on identifying clusters of patients by using unsupervised methods. Due to regression to the mean, patients with greater baseline diseases are more likely to be identified as good responders, while those with milder diseases are more likely to be identified as poor responders. In fact, the response is actually a continuous spectrum.

6. Finally, I feel a little uneasy that equating the relation between change and the baseline values to the proportional recovery rule. I feel the latter is more akin to the percentage change. Please see the paper: Testing the relation between percentage change and baseline value. Sci Rep 6, 23247 (2016).

I think the authors made great efforts to clarify various misconceptions about testing the relation between change and the baseline. To get your messages across the intended readers, I suggest making your discussion more focused. For example, I do not feel that introducing bootstrapping or GAM is necessary. In contrast, the key concepts of mathematical coupling and regression to the mean were not explained in the paper. In my experience, they are among the most different concepts to comprehend even for statisticians.

*Reviewer #3 (Recommendations for the authors):*

This is a defence of the Proportional Recovery Rule (PRR), which asserts that stroke survivors recover a fixed proportion of lost function after stroke, from recent, formal/statistical inspired criticisms. The analyses and data show that the PRR is likely to be relevant to ongoing efforts to understand and model post-stroke recovery, and the initial severity of upper limb motor impairments (hemiparesis) after stroke predicts 35-55% of the variance in their subsequent recovery from those deficits.

This manuscript includes a lot of careful analyses that seem reasonable to me and are novel as far as I know in this specific domain. The authors' conclusions also appear to be supported by their methods and data – and their data are impressive, with three large, relevant datasets. I note that other reviewers have identified some detailed issues with the analysis and the text, but in the main the authors have done a very good job of addressing those concerns. The one exception, in my view at least, concerns the best strategy for dealing with patients at or near ceiling in the first two weeks post-stroke. This issue was raised by another reviewer, with whom I co-authored a paper outlining our favoured response, which is to exclude these patients from analyses seeking to quantify the explanatory power of the PRR (Bonkhoff et al., 2021, JNNP). By failing to exclude those acute-near-ceiling patients, the authors have left me somewhat sceptical of the reliability of the variances explained that they report. But this is perhaps more a matter of taste than of statistical rigour, and I can certainly appreciate the authors' reluctance to exclude hard-won empirical data. In their place, I might reference this issue as a possible limitation in the manuscript.

In other words, this paper is not 'broken' in my view, so I have no objection to its publication – and no real requirements for revisions.

That said, I also do not believe that this paper makes a particularly compelling contribution to the field: it 'defeats' a straw man caricature of the criticisms made of the PRR, and offers evidence in support of a position with which even the rule's most critical commentators already agree (and indeed for which some of them have already published supporting evidence).

The original criticism of the PRR was that the analyses used to support it would yield apparently enormous effect sizes entirely regardless of whether the PRR was really relevant to recovery (Hope et al., 2019, Brain). There was never any assertion that the PRR was irrelevant to recovery: that conclusion could never have been justified merely from the recognition that the empirical support for the PRR was unreliable. A follow-up analysis with a large dataset (albeit not as large as that used here) suggested that the PRR was indeed really relevant to recovery, but that it explained a less of the variance in recovery than prior empirical analyses had suggested (Bonkhoff et al., 2020, Brain). That latter work shares much in common with the analyses presented here: i.e., it is a model comparison analysis, with the PRR as one of the considered models. The current work is also a model comparison analysis, with the PRR as one of the considered models, but implemented with different methods and tested against a lot more data. In other words, this paper is in my view a conceptual replication of the earlier study: using different methods and data to run a broadly similar analysis, which yields broadly similar conclusions to those reported previously. Similarly, while the lengthy discussion of clustering is well done, it adds little (in my view) to the conclusions already drawn in Bonkhoff et al., 2022, JNNP.

In other words, much of what this manuscript claims to prove, in response to the critics, has already been reported by some of those same critics. In this sense, the paper is akin to a conceptual replication; using excellent data and novel methods (for the domain at least) to draw conclusions that converge with what has come before, and expressing it all in an accessible manner. These are all strengths in my view: I would merely ask that the links to prior results be made more explicit, at least so that others can follow the timeline of the debate more easily. Or indeed if it's the authors' view that I am wrong, that this be justified more explicitly.

Finally, I am concerned that this issue is rather too narrow to appeal to the readership of *eLife*. To bolster the more general appeal, I would recommend adding a paragraph or two on variants of this debate that have raged in other subjects – examples of which the authors themselves have given in other papers on this topic.

---

## [Author Response]

[Editors’ note: the authors resubmitted a revised version of the paper for consideration. What follows is the authors’ response to the first round of review.]

Reviewer #2 (Recommendations for the authors):This paper addresses the inherent potential of motor recovery after stroke. The authors would like to validate the concept of proportional recovery which assumes that stroke patient typically show a fixed amount of recovery of about 70% irrespective of where they start. The authors suggest that after controlling for possibly confounding variables like ceiling effects or mathematical coupling, they still could validate the proportional recovery rule. The Results section is difficult to read for a non-statistician. However, the findings reported in the paper are rather incremental than novel. Most importantly, there are no data that explain the neurobiological basis of proportional recovery. Hence, the novelty of this submission remains low apart from more sophisticated statistical models which ultimately come to a similar conclusion than reported years before.

We appreciate the time taken by the reviewers to evaluate our work but disagree that the results presented here are incremental.

Proportional recovery has shaped thought about recovery from stroke across several domains for nearly a decade; recently, the statistical foundations of work focused on the PRR have been critiqued from a variety of perspectives. Our goal is not necessarily to “validate” the PRR, but to provide a comprehensive rejoinder to critiques that contain some merit but overstate their conclusions and miss important insights. We have made efforts to provide this in a way that is accessible, recognizing that the arguments are undoubtedly nuanced, and have suggested several novel techniques for assessing a range of statistical and mechanistic models for recovery. We have proposed new tools for evaluating hypotheses regarding the existence of multiple recovery groups. Finally, we have used these tools to reevaluate existing data in light of the recent criticisms by testing alternative models for recovery.

Given the current discourse around the PRR, it is critical that this perspective be available to interested readers. Otherwise, it is likely that the field might conclude the PRR has been invalidated or lacks a robust statistical framework. Neither is true; our results (and, indeed, frequently the results of those offering critiques) suggest that the PRR is among the best predictive models for recovery currently available.

As stated above, the Results section is very difficult to read and appears somewhat lengthy. Lesion analyses or any other data about the possible neurobiological foundations of the proportional recovery are missing. I acknowledge the relatively large sample size, but the Fugl-Meyer score based view without any further para-clinical evidence is somewhat meager. Therefore, I do not see that this paper has a major impact on the field, as in the best case it confirms the proportional recovery rule which has been published many years before.

We appreciate this feedback. Specifically, our goal in this work is not to produce new clinical evidence or develop a new model framework. Instead, we collate and address the range of statistical critiques of the PRR that have appeared in recent years. We disagree that our contribution is support a rule that has been published and studied, and therefore is incremental at best; rather, this manuscript introduces a range of novel statistical approaches that can used to understand results related to recovery. This nuanced statistical understanding is relevant beyond any specific argument regarding the PRR, because relating baseline predictors to follow-up values is a fundamental issue in clinical practice and scientific discovery.

Reviewer #3 (Recommendations for the authors):This paper presents a defence of the proportional recovery theory of how stroke patients with upper limb motor impairments recover. This is a theory that has come under attack due to confounds associated with the analyses used to assess the theory. The paper acknowledges these confounds, which include compression (to ceiling) enhanced mathematical coupling and over-fitting in the differentiation of recovers and non-recovers. The paper then presents a number of methods: use of non-zero nulls; focus on the variance-ratio; bootstrapping to assess variability in parameter and model fits; comparison of model fits; etc. These certainly improve upon the methods classically used in the literature to justify the proportional recovery hypothesis. Additionally, we appreciate the constructive tone of this paper and its effort to arrive at a consensus position on the proportional recovery issue. Indeed, the overall conclusion that there is a proportional (to lost) recovery pattern amongst the moderately impaired group is a position that we have sympathy with, although that pattern is much weaker than some previous claims in the literature.

Thank you for the careful reading of our work. Your assessment of our contributions here – new methods that improve on those that have typically been used to evaluate the PRR and other models for recovery – is in line with our own. As you note, our goal is to be constructive: there are a number of valid critiques of traditional methods, and our work arose after careful consideration of recent criticisms. And it seems we agree (to some extent) about the usefulness of proportional recovery for at least some groups, as well as about the issues in previously reported effect sizes and significances.

However, there seem to be remaining problems with the analyses employed, although, it is difficult to definitively judge the methods employed without more detail than the paper currently offers.The two most important issues that seem to be weaknesses are as follows.1) Remaining confounds with key measures: Goldsmith et al. argue that particular combinations of correlation between baseline and follow-up, i.e. cor(x,y); correlation between baseline and change, i.e. cor(x,δ); and the variance-ratio, i.e. k, is diagnostic of proportional recovery (PPR). In particular, they argue that the combination of cor(x,y), cor(x,δ) and k shown in table 1 and figure 5 justify their claim that the three datasets they consider, Stinear and Byblow, Winters and Zarahn, definitively exhibit a PPR pattern. However, the simulations presented in Bonkhoff et al. (2020), show that this same combination of measures can be exhibited by a constant recovery model in the presence of ceiling. Indeed, this is the fundamental confound described by Hope et al., Bonkhoff et al. and Bowman et al., which led to the conclusion that the only way the ceiling confound could be avoided is by throwing data points at ceiling away. (This is actually what I believe you should do with the data in this paper. Note, in Bonkhoff et al. (2020), we verified with model recovery simulations that the throwing away procedure enables the correct model to be recovered.) To make this completely clear, I have prepared a figure, which I am assured will be attached with this review.https://submit.elifesciences.org/eLife_files/2021/09/17/00098237/00/98237_0_attach_15_32737_convrt.pdfThis shows table 1 and figure 5 from Goldsmith et al. on the left and figure 6 from Bonkhoff et al. on the right. I have added annotations to figure 6, which show where the Stinear and Byblow and Winters data sets would sit in the panels for constant recovery. The Zarahn data set does not so obviously correspond to a vertical line, but fits with the general combination of measurements.

The statistical issues surrounding the PRR often begin with concerns about mathematical coupling – correlations that can be induced when relating baseline to change scores, even when baseline and follow-up are independent. Additionally, it is possible that observing different outcome variances at baseline and follow-up can produce stronger than correlations between baseline and change, regardless of the correlation between baseline and follow-up. We found it necessary to discuss the interdependency of cor(x,y), cor(x, δ) and k as a way of introducing important statistical concepts, providing groundwork for some of our proposed techniques, and differentiating between “mathematical coupling” and the effects of compressing the outcome distribution. These concepts are revisited when we note many similar issues arise even when alternative statistical methods, including mixed models, are used.

That said, we do not intend these metrics to be “definitive” or even “diagnostic” in support of the PRR. Indeed, we are careful to say that multiple mechanisms could give rise to these observations, including constant recovery with a strong ceiling effect. We will comment in more detail on our suggestions for comparing methods in our response to your next point, including methods for addressing the ceiling confound. We have also reiterated that the correlations and variance ratio is not intended as definitive in our revised work.

The Figure you provided is a useful way to frame results. We are not surprised that one could find correlations and variance ratios similar to those found in Byblow and Stinear and Winters et al. through constant recovery and a ceiling effect. However, this figure suggests a constant recovery of 35 and 45 points in those respective studies. That would suggest that nearly all mildly and moderately affected stroke patients are expected to recover to ceiling, which is not consistent with the data or our experience. Meanwhile, we note that one could add similar annotations to the “Proportional to Lost” column in Bonkhoff et al. 2020, and conclude a recovery proportion of roughly 60% and 80% for Byblow and Stinear and Winters et al., respectively.

2) Limitations of model fits: certainly, the effort to fit models to the three data sets is welcome. However, there is a concern that this model comparison is not a completely fair test of competing theories to proportional recovery. The model fitting comparison in figure 6 is interesting to see. Although, I do find it difficult to interpret the findings, for the following reasons: (a) a criterion has not been given to judge when one model is, in a statistical sense, doing better than another model. I would have expected an assessment of where the MAPE for the best model sits in the bootstrap distribution of MAPEs for each of the other models. This would enable an inference of the kind that the MAPE of the best model is beyond the 95% confidence level of the second best model, and similar for the third. Something of this kind was done in Bonkhoff et al. (2020). (b) I suspect the reason that the Generalised Additive Model does not win is because it is too flexible. It would have been very useful to have seen (perhaps in an appendix) the exact functional form of the Generalised Additive Model. So, I cannot be sure, but I suspect it was not set up to be specifically for saturation at ceiling patterns, with a positive slope leading to a saturation on the right. Consequently, I suspect that the model overfits for many of the bootstrap samples. van der Vliet et al. (2020) give a strong precedent for using a simple exponential, which I suspect would fit the data much better. (c) when I look at figure 4, left panels, lower row, I feel that I do see a strong ceiling pattern in the data. However, this would be much easier to assess if simple x against y (baseline against followup) scatter plots were also presented and model fits for each model were depicted on the scatter plot. This could for example go into an appendix.van der Vliet, R., Selles, R. W., Andrinopoulou, E. R., Nijland, R., Ribbers, G. M., Frens, M. A., … and Kwakkel, G. (2020). Predicting upper limb motor impairment recovery after stroke: a mixture model. Annals of neurology, 87(3), 383-393.

The discussion of model comparisons is critical because, as we noted in response to your previous comment, we intend this as a more helpful metric for distinguishing between models for recovery than an examination of correlations and the variance ratio. We agree that trend toward explicit model comparison is helpful, and respond to your three points below.

a)We have not provided a statistical metric that allows claims of statistical significance because these are generally not available. Cross validation provides a way to compare prediction accuracy, but does not examine a model parameter or pose a statistical hypothesis to be tested; confidence intervals or p-values for comparing models are therefore not available from this procedure. Instead, the results provide an understanding of the distribution of prediction errors that can guide discussions of model fit. Often, when multiple models provide similar fits, the most parsimonious model is preferred. Our results suggest that the distribution of prediction errors for several models are similar, and we suggest the PRR as a simple model that is comparable (or perhaps slightly better than) more flexible fits.

b) Our implementation of the GAM uses the mgcv package in R, which is one of the best known tools for modeling non-linear associations. The general modeling framework is described in “Generalized additive models: an introduction with R”, and the software contains many methodological and computational improvements since that book was written. mcgv uses a rich b-spline basis with explicit smoothness penalties to prevent overfitting. We include a more complete specification in the appendix, and show a random sample of GAM model fits in CV splits for each dataset. The GAM includes the PRR as a special case, and so in some sense the worse performance is because fits are “too flexible” – but this stems from minor fluctuations rather than dramatic overfitting, and the GAM fits are often similar to those obtained from the PRR.

The GAM model does not specifically introduce a saturation point or a positive slope. Van der Vliet suggests an exponential model for long-term recovery trajectories rather than as a model relating baseline to follow-up, but this is a realistic parametric form and we have included it in updated results.

Additionally, constant recovery with a ceiling effect of the form y=min{x+c+ϵ, 66} can be viewed as a special case of Tobit regression in which the only free parameter is the constant c. Tobit regression includes all data points, including those at ceiling, when estimating model parameters, but assumes a latent observation that follows the specified linear form. We have implemented this special case of Tobit regression and included it as a competing model.

Updated model comparison results are included in the revised manuscript. In short, constant recovery with a ceiling substantially underperforms against competing methods. The GAM, exponential, and PRR prediction accuracies are often comparable, although the PRR appears superior for the Stinear and Byblow and Zarahn datasets. These results more completely substantiate our claim that “competing models for recovery, specifically, a non-linear association between baseline and follow-up, which might arise from near-constant recovery in the presence of a ceiling effect, does not produce more accurate predictions than the PRR.”

c) We have added the suggested figure, to the appendix. This shows fits produced for the training datasets in our CV procedure for each model. In keeping with the results for prediction accuracy, the constant recovery model is visually a poor fit for these datasets. In some instances, the GAM is indeed too flexible, especially for the Zarahn data. That said, the GAM and exponential models are – unsurprisingly, given the results for prediction accuracy – similar to the PRR in terms of fitted values despite their additional flexibility and complexity.

I have the following more specific comments for the authors.Abstract: the following statement is made," We describe approaches that can assess associations between baseline and changes from baseline while avoiding artifacts either due to mathematical coupling or regression to the mean due to measurement error". For reasons justified above, and below, I am doubtful that this statement and similar statements made elsewhere in this paper are appropriate.

In our response to previous points and your comments below, we argue that this claim can be made. Meanwhile, we have attempted to be more clear that some tools for assessing association, especially those based on correlations and the variance ratio, are sensitive to ceiling effects.

Introduction section, paragraph beginning "The growing meta-literature discusses …", statement: "the nuanced and sometimes counterintuitive statistical arguments are critical to get right for the sake of furthering our understanding of the biological mechanisms of recovery." I could not agree more than I do with this statement.

As elsewhere, it’s welcome to have points of agreement.

Results section, paragraph beginning "A broader argument relates to settings..", with regard to the reference to Bonkhoff et al. (2020) in this paragraph, the reference to "degenerate" is actually most explicitly made in Bowman et al. (2021).

Thank you for noting this. We have updated the reference as suggested.

Results section, subsection "Distinguishing true and artifactual signals", paragraph beginning "The value of cor(x, δ) depends on the variance ratio κ and..", statement: "The variance ratio can be used as a measure of the extent to which recovery depends on baseline values, regardless of the value of cor(x, y)." As indicated in our first point in the Public Review section, if this statement is suggesting that the variance-ratio can be used to unambiguously identify a proportional recovery pattern, we do not agree. Indeed, the variance-ratio could be small even when there is no recovery at all; that is, if the mean at follow-up is the same, or indeed below, the mean at baseline.

We have edited this statement to reiterate your point (and ours), namely that the variance ratio can be small under a variety of data generating mechanisms.

Results section, subsection "Distinguishing true and artifactual signals", paragraph beginning "All datasets have x values generated from a Normal distribution…", how is ceiling handled in these simulations – I would have expected that sometimes a followup score there would be, "by chance", above 66. Can you add a explanation of what happens to these?

In these simulations, we initialized random number generation using seed values that produced no observations above 66 or below 0. We made this choice because our focus here was on issues that arise from mathematical coupling and small variance ratios, while ceiling effects as a possible driver of the variance ratio is deferred to later sections. However, the same conclusions can be drawn for datasets in which ceilings and floors are explicitly enforced.

Results section, subsection "Distinguishing true and artifactual signals", a minor frustration is that the contour plot on the right of figure 1 is the opposite way around to the surface plot in Hope et al., with a baseline by follow-up correlation of one furthest to the right. At the least, could this discrepancy be mentioned in the caption to help the reader.

Our formulation of Equation (1) has cor(*x*, δ) as a function of cor(*x*, y) and k; we therefore prefer to have cor(*x*, δ) on the y axis and cor(x, y) on the x axis in Figure 1. We have included your suggested note of the discrepancy to orient readers in comparing figures.

Results section, subsection "Distinguishing true and artifactual signals", para beginning "Dataset A is a canonical example of mathematical coupling", I am unsure about quite a lot of the points made in this paragraph. It seems to me that a lot of what is said in this paragraph does not sit well with point 1.

In this paragraph and elsewhere, we have tried to clarify that our points in this section are intended not as definitive evidence of the PRR and acknowledge that other mechanisms can give rise to correlations and variance ratios similar to those obtained when data are in fact generated by the PRR. Indeed, in this section we do not use the PRR to simulate data, instead focusing on the relationship between of cor(x,y), k, and cor(x,δ). We maintain that beginning with these statistical observations is important for understanding later arguments, and in fact are not in conflict with many of the recent critiques of the PRR.

Results section, subsection "Distinguishing true and artifactual signals", para beginning "Dataset D represents the least controversial..": I would agree that dataset D is a clear example of proportional recovery. However, our central point in previous publications is that on the basis of the methods typically used, including in this paper, that judgment has to be made informally on the basis of visual inspection. This is because one can obtain exactly the same values of statistical measures (r(X,Y), r(x,δ), k), in a dataset without PRR, but which contains a strong ceiling effect, and indeed, datasets do typically exhibit strong ceiling effects; that is, from what I have seen, empirically collected datasets do not typically look like dataset D, because of the ceiling effect.

There has been a broad collection of publications critical of the PRR in recent years, some by the referee and co-authors and some by other groups. Often these begin with a straightforward discussion of mathematical coupling, and have moved on to critiques that involve a reduction in outcome, clustering, and other statistical issues. Our goal in this manuscript is to provide a broad reflection on and rejoinder to these critiques; we felt it necessary to begin with mathematical coupling, correlations, and variance ratios, but in this section explicitly note that several mechanisms can give rise to the same observed values. We introduce simulated datasets primarily as illustrations, and in later sections propose methods to distinguish between mechanisms that underlie observed data.

Please note as well that we referred to this as the least controversial among the four datasets we presented, rather than as an example of the PRR, because it is a setting where both cor(x,y) and cor(x,δ) are large.

Results section, subsection "Distinguishing true and artifactual signals", para beginning "Taken together, the..", text: "We also identify a setting, typified by Dataset D, in which each measure suggests the presence of a relevant association. It is not the case that data like these necessarily imply that recovery follows the PRR. Other biological models could produce data similar to Dataset D, and how to compare competing models will be considered in later sections." Just to say, I was a little confused here. What you are saying here seems to be inconsistent with what you said in the previous paragraph in re. what can be deduced from dataset D.

We have attempted to clarify the text throughout this section in response to your comments, including this one. This text was intended to convey a sentiment it seems you agree with – namely that the PRR can give rise to a collection of correlations and variance ratios, but that other mechanisms can as well. Tools in later sections are intended to differentiate between these, and our purpose in this section and elsewhere is to provide relevant background and our perspective on the statistical issues that have been raise elsewhere.

Results section, subsection "Recasting Oldham's method", paragraph beginning "Instead of cor(x+ γ, x − γ), we prefer to focus…": again, some of what is said in this paragraph seems to ignore the confounds that arise from ceiling effects.Results section, subsection "Correlations, variance ratios, and regression", paragraph beginning "The following expressions relate..", Would it be possible to see a derivation of these two equations? These are not completely standard since \δ = y-x and ii = max-x. This could go in an appendix.

We have broadly revised our text to acknowledge that ceiling confounds can give rise to correlations and variance ratios similar to those that are produce by the PRR. Meanwhile, in our original submission this paragraph included the following text:

“For instance, as noted by both (Hope et al. 2018) and (Hawe, Scott, and Dukelow 2019), Oldham’s method does not address the possibility of ceiling effects; in our view, determining which process gives rise to observed correlations comes after assessing whether those correlations are artifacts driven by coupling.”

This was intended specifically to address this point.

Results section, subsection "Comparing distinct biological models for recovery", paragraph beginning "To illustrate how prediction accuracy…", last sentence: can you give more details of how ceiling is enforced?

We have revised the text in this sentence as requested. In short, data above the ceiling are set to the ceiling value. We note that in our first submission data at ceiling were taken to be the maximum value minus an error generated from an exponential distribution, while in our current submission we generate data under a strict ceiling effect.

Results section, subsection "Comparing distinct biological models for recovery", paragraph beginning "We consider three models for the association …", you say that "Second, we implement the PRR (without intercept) to estimate δ given *x*, with *y* taken to be *x* + δ.", Don't you need to tell us the slope in this eqn, in order to know it is proportional to lost? This is required to rule out proportional to spared or constant recovery. Also, you refer to using a "generalized additive model". Could you give more details of the functional form of this model, perhaps in an appendix? At the least, could you include a relevant reference?

Our implementation of the PRR is, we believe, described as “proportional to lost” in Bonkhoff et al. 2020 and elsewhere, and we have made that explicit in our revisions. We have also clarified that we estimate the proportion based on observed data.

As noted in response to part (b) of your second major concern, we use the framework introduced in “Generalized additive models: an introduction with R” and implemented in the mgcv package. This estimates non-linear associations using a flexible spline expansion with explicit penalties to enforce smoothness in a data-driven way. We have included these details and references in our revisions.

Results section, subsection "Results for reported datasets", paragraph beginning "We next conducted the bootstrap analysis on the subsample…": again, the confound created by ceiling effects impacts this paragraph.

Our conclusion in this paragraph is “we have evidence both that recovery is related to baseline values in a way that reduces variance at follow-up, and also that baseline values are predictive of follow-up” but not that a particular mechanism drives this finding. As elsewhere, we have clarified that several data generating mechanisms could give rise to results like this, and suggest alternatives for distinguishing between those mechanisms.

Meanwhile, we think these results (and the discussion of correlations and variance ratios more broadly) are important for understanding issues that arise when relating baseline, follow-up, and changes scores. In particular, these help to contextualize observed correlations and address concerns about canonical forms of mathematical coupling.

Results section, subsection "Results for reported datasets", paragraph beginning "We compared the performance of three models …": this is where my second point above applies.

As noted in our response above, we have substantially edited this section. The main changes are a clearer interpretation of the meaning of CV results; comparison to additional models for recovery, including constant recovery with a ceiling effect (implemented as a special case of Tobit regression) and an exponential model; and a more complete discussion of competing approaches.

Results section, subsection "Results for reported datasets", table 2 caption. Sorry for being dumb, but could you give more explanation of how the R-squared is being calculated here. In particular, could you confirm whether the R-squared equation given in the "Correlations, variance ratios, and regression" subsection is the one being applied here. If it is not this one, can you give the equation that is being used. This is a key issue, since the R-squared values here are much smaller than those given in relevant papers published for some of these data sets.

Thank you for noting this inconsistency. In “Correlations, variance ratios, and regression” we derive an analytical form for R^2^ when regressing δ on x and show that this is cor(x,δ)2, as part of a larger discussion of how large correlations and R^2^ values are expected in certain circumstances. In “ Results for reported datasets”, however, we are interested in understanding variability in the outcome y, and how much can be explained by baseline values x. The R^2^, like the MAPE, measures the strength of that association. We have updated our text to clarify this distinction.

Results section, subsection "Results for reported datasets", paragraph beginning "The preceding results for …": as previously discussed, the central finding of the work of Hope et al. and Bonkhoff et al. is that, in the presence of ceiling, recovery patterns completely different to the PRR, for example proportional to spared function, can look like PRR, when the measures considered in this paper, cor(\δ, x), cor(x,y), var ratio, are taken. How has the work presented here countered that criticism? I cannot see that it has.

We have edited this paragraph to more carefully describe our results and conclusions, and made similar edits elsewhere in the paper.

We agree that similar correlations and variance ratios can arise through the PRR, constant recovery and a ceiling, proportional to spared function, or other mechanisms, and our goal is not to counter that criticism in Hope et al. or Bonkhoff et al. Correlations have often been used to measure association in studies of recovery, but can have very counterintuitive properties. Our discussion and results show that some common statistical issues – like a canonical form of coupling, which is a common example in papers critiquing the PRR – can be ruled out using appropriate “null” values.

Meanwhile, the cross validation results that are also included in this paragraph can form a basis of comparison for competing models. It is in reference to these results that we claim “competing model for recovery … do not produce more accurate predictions than the PRR”, which is supported by our findings.

Discussion section, paragraph beginning "Indeed, a recent large-scale cohort study …", it is stated that, "Where Hope et al. (2018) and Bowman et al. (2021) note ceiling effects can compress scores in a way that induces a variance reduction, Lee et al. (2021) observe more than half of all patients, and more than 60% of patients with baseline FMA-UE above 46, recover to the maximum possible value." I do not understand what is being said here. Lee et al. is put in opposition to Hope et al. and Bowman et al., but isn't the Lee et al. ceiling effect going to exactly lead to a reduction in variance at followup?

We phrased this poorly, and have attempted to clarify our meaning in the revised manuscript. Our intended point was that Hope et al. (2018) and Bowman et al. (2021) are largely focused on the impact ceiling effects can have on observed correlations, while Lee et al. (2021) model the probability of full recovery as a function of initial impairment and other variables using a logistic regression. We believe that evaluating the data presented in Lee et al. (2021) using a range of competing models to evaluate their predictive accuracy would be a useful contribution.

Discussion section, paragraph beginning "More complex models than the PRR do.." What is being said in this paragraph about Bowman et al. (2020) is not completely clear to me and this may well be a presentational failure on our part when writing Bowman et al. The only place in Bowman et al. (2020) where we refer to mixed models is the following statement, "In particular, van der Vliet et al. present an impressive Bayesian mixture modeling of Fugl-Meyer upper extremity measurements following stroke. Importantly, the authors avoid the confounded correlation of initial scores with change by simply fitting their models to behavioral time-series, that is, raw Fugl-Meyer upper extremity scores, without involving the recovery measure."However, in the next paragraph, you commend van der Vliet et al. for the same piece of work that we are referring to. Indeed, we were under the impression that the above quoted statement in Bowman et al. (2020), was only a reiteration of what van der Vliet et al. say themselves, which is as follows, "Our current longitudinal mixture model of FM-UE recovery, as opposed to the proportional recovery model, cannot be confounded by mathematical coupling. Hope et al. showed that the correlations between baseline FMUE score (distribution X) and the amount of recovery defined as endpoint FM-UE minus baseline FM-UE (distribution Y-X) found in proportional recovery research could be inflated by mathematical coupling. However, because mathematical coupling applies to correlations of data points (baseline and endpoint FM-UE) and not to models of longitudinal data, the recovery coefficients in our research represent nonconfounded measures of recovery as a proportion of potential recovery. In addition, mathematical coupling does not apply to the outcomes of the cross-validation, as we report correlations between the model predictions and the observed values for endpoint FM-UE and ΔFM-UE rather than correlations of the form X and Y-X."This leaves us confused, as to what about Bowman et al. (2020) is being criticised in your paragraph beginning "More complex models ….".

Our attribution of this claim – that longitudinal models avoid the statistical issues inherent in relating baseline and change scores by posing a model for outcomes y rather than change scores – to Bowman et al. (2021) and Lohse et al. (2021) was incorrect. We apologize for the oversight and have corrected the mistake.

The point made in that paragraph is another example of ways in which statistical intuition can fail. It is reasonable to think that a longitudinal model “does indeed avoid mathematical coupling since the change variable plays no part”, as it is put in Bowman et al. (2021). But if one uses a mixed model for the canonical example of mathematical coupling, the resulting random intercepts and slopes will have a correlation of -0.717 – exactly the same as the counter-intuitive correlation between X and Y. Put differently, the concerns about correlations, variance ratios, and model comparisons persist even when data and models are more complex, and may become even harder to spot.

[Editors’ note: further revisions were suggested prior to acceptance, as described below.]

The manuscript has been improved but there are some remaining issues that need to be addressed for further consideration, as outlined below:Essential revisions:1) Please expand your discussion of the similarities between the current work and previous critiques of the PRR. This should include a brief comment on the justification of some of the analysis choices (e.g. inclusion/exclusion of patients at ceiling) and the relevance of the findings presented here to other areas of research.

In our revisions we have more carefully tracked the critiques of the PRR and the more recent convergence in the field. As part of this, we note similarities and differences in analysis approaches and include justifications for our model choices. Finally, to broaden the appeal of this manuscript to *eLife*’s readership, we comment on the relevance of this work to other domains.

2) Address the statistical concerns raised by Reviewer #2.

We value input from this reviewer, who has clear expertise in this area, and appreciate that the reviewer recognizes our “great efforts to clarify various misconceptions about testing the relation between change and the baseline.” We agree with the majority of points raised in these comments, and have revised accordingly; please see our responses below for details.

3) Based on some of the reviewers' comments, you may decide to move some of the analyses to a supplement. This is at your discretion.

We appreciate the reviewers’ suggestion to focus on succinctly introducing the statistical issues and our contributions, and to avoid unnecessary detail or digressions. At the same time, a careful consideration of the PRR, its critiques, and our insights requires some nuance and detail. In response to this comment and those from the reviewers, we have deferred some discussions to the appendix and proofread the manuscript for conciseness.

Reviewer #1 (Recommendations for the authors):The authors have completed an extensive reworking of their paper, which does a good job of responding to the majority of my comments. Perhaps most importantly, the authors have convincingly countered the argument that I made in my first review that their findings could have arisen from a Constant model with a ceiling. It is excellent to see this rebuttal of my concern. So, I am basically happy with this revision.It is again good to see the emphasis in this paper on model comparison – that certainly seems to be the right way to go. I also note that the findings in this paper are somewhat consistent with what we have reported in Bonkhoff et al., 2020 – a PRR effect, although considerably weaker than had previously been claimed. So, as the authors point out, there does seem to be convergence in the field, which is excellent.

The first review of our work was thoughtful, and we believe the careful revisions undertaken to address the comments raised at that time improved the quality of our work. The inclusion of constant recovery in the presence of a ceiling effect is, as you note, an important element of the revised manuscript: the referee’s prior work illustrated that this data generating mechanisms could result in findings consistent with the PRR, and explicit model comparisons in observed data was warranted.

We’re happy to see the growing convergence in the field around a number of points, and continue to highlight these in our revised manuscript. We further agree that careful model comparisons are critical for understanding statistical models and biological mechanisms of recovery, and that the results of our analyses are broadly consistent with those reported in Bonkhoff et al. 2020. Meanwhile, we would argue that the contributions of this manuscript are broader than model comparisons or validations of previously reported results; please see our response to the third reviewer for a more complete discussion of this point.

It remains a pity that the claim that PRR is definitively better on the basis of the presented model fits is still a weak claim. In particular, violin plots for GAM, Exp and PRR are heavily overlapping in figure 6. As you surely know, what is needed is a more sophisticated model comparison, most likely of a Bayesian variety, using something like the model evidence, free energy or deviance information criterion.The key inference suggested in the paper hinges on an Occams razor argument that if one cannot distinguish between GAM, Exp and PRR, one should pick PRR, since it is more simple. [Although note, I do not believe that Exp is a supermodel of PRR, i.e. I don't think one can get the PRR pattern from Exp, which slightly complicates this line of argument.]I think that in the absence of a more sophisticated model comparison, of the kind I just highlighted, the Occams razor argument probably is the best one can do. However, since the cross validation is out-of-sample, the flexibility of the model is actually taken into account there.

We use cross validation to compare models because it explicitly focuses on prediction accuracy, and the results can therefore be interpreted in the context of single-subject predictions. This implicitly takes model complexity into account by separating the training and testing datasets.

The more formal approaches you list here and their application in Bonkhoff et al. (2020), Bonkhoff et al. (2022), and others, are valuable contributions to this literature. They explicitly balance goodness-of-fit with model complexity, and the Bayesian framework provides a complete assessment of all forms of model uncertainty. In our revisions, we emphasize the importance of these tools for model comparison and highlight their use.

The broadly overlapping violin plots suggest that models achieve similar prediction accuracies, which makes model selection difficult. They are reminiscent of the deviances shown in Figure 7E of Bonkhoff et al. 2022, where Standard Form Regression and Proportional to Lost have very similar performances. Even with sophisticated model comparisons, settings where data are not sufficient to identify an obviously preferred model will exist, with resulting ambiguity for model selection and inference.

Among the models we consider, the PRR is simplest – it is linear model controlled by a single, interpretable parameter. This simplicity and interpretability is one reason we favor the PRR over competing models, but the examination of fitted values suggested in the first review is another. If more complex non-linear models yielded similar prediction accuracies but obviously different fitted values, that might suggest competing biological mechanisms are equally well supported by the data. The PRR is not a special case of the exponential we’ve implemented, but the fits of the PRR, GAM, and exponentials models are similar enough that we favor the simpler and more easily interpretable PRR. As an aside, if two models produced with distinct fitted values produced similar prediction accuracies, leveraging the unique information they provide via Bayesian model averaging or ensemble prediction could be explored.

Obviously, it would be great if this issue could be resolved with a more sophisticated model comparison in a further re-write, however, I don't think this is fair to require at this stage of the review process, but it would be good to acknowledge this weakness in the discussion, and note that competitor approaches are doing better in this respect.

We see value in pursuing sophisticated model comparisons that include a range of candidate models in future work, and appreciate that the review agrees that work may not be necessary for our contributions at this stage. We have expanded our discussion of this point and acknowledged prior work that uses more sophisticated techniques for model comparison.

Specific points:I did not pick this up in the first review, but I spent a lot of time on this review being confused by the right panel of figure 1, which re-represents the surface plot from Hope et al. (2018). I'm now pretty sure your colour bar needs to be inverted. As it is, the yellow contour corresponds to the degenerate region in Hope et al's surface plot, i.e. it does not matter what the value of cor(x,y) is, you always get a correlation of x with change near to -1. However, shouldn't that be for low values of the variability ratio (i.e. log(k) substantially smaller than zero)? Your yellow contour corresponds to log(k) much *bigger* than zero. This issue is also present in Figures 4 and 5.

You are correct, and we appreciate you noting this issue. We have corrected our underlying plot functions and resolved the inversion of the color bar in all plots.

Top of page 8: "which when baseline (x) and follow-up (y) are uncorrelated and have the same variance." Should the "when" be deleted here?Page 9, towards bottom: "relationship between of".Page 10, 2^nd^ para: “data represent in which baseline values”.

Thanks for noting these language mistakes; we have corrected these and proofread the rest of the paper.

Page 17, beginning of last para: you talk about there being five models, but in the next sentence you only list 4.

You are correct; our list had omitted the exponential model motivated by van der Vliet. This is corrected here and on page 38, noted below.

Page 26: you reference Lohse et al. 2021, but I couldn't find this in the bibliography.

This entry in the bibliograph was appended to the preceding item, Lee et al. (2021), rather than appearing as a new citation. We have fixed this issue in the revision.

Page 38 top of page: there are a couple of typos here. Also, you again talk about there being five models, but in the next sentence you only list 4.

We have fixed these typos, corrected the list here as well.

Reviewer #2 (Recommendations for the authors):1. I think this paper could be made more succinct. It is quite long because the authors used many examples to demonstrate their arguments. Although I feel this approach is well-suited to clinical audience, I am not sure whether the targeted readers may get lost in the technical discussion. For example, I am not sure using bootstrapping is necessary, as we can either directly test the spurious correlation against a more appropriate null hypothesis or using simulation (see the paper in Eur J Oral Sci 2005; 113: 279-288).

As the reviewer is aware, there is an inherent challenge in making the nuanced statistical issues that arise in this setting and their solutions accessible. Our manuscript also synthesizes and contributes to an ongoing discussion in this literature; we provide necessary background to frame our discussions and reinterpret some prior examples in light of our perspective. Throughout, we strive to balance informative examples with a need to be succinct.

We agree that direct testing is possible when sample sizes are large and other conditions are met, and have emphasized this in our revision. Simulations, as presented in the paper you note, provide a mechanism for obtaining null values when sample sizes are small (as they often are both in the periodontal studies that motivate that paper and in studies of stroke recovery). The simulation approach depends on generating and then drawing repeated samples from paired data with an observed correlation; like the bootstrap, this is an inherently iterative procedure. While both are plausible, it is not obvious to us that one is more accessible that the other and we prefer to retain the bootstrap for constructing confidence intervals.

Your broader point about brevity is well-taken. In particular, in response to concerns you raise below (which echo broader critiques in this literature), we have moved the digression on R^2^ and regression coefficients to a supplement. We have also edited our manuscript with an eye to removing unnecessary components.

2. There are two major issues involving in assessing the relation between change and the baseline. The first is the hypothesis testing and the other is the prediction. Because of mathematical coupling, the usual null hypothesis that the correlation coefficient is zero is inappropriate. This is because the distribution of correlation coefficient is in a restricted space defined by the correlation between x and y, as shown in Figure 1. By the way, I think the paper by Bartko, J. J. and Pettigrew, K. D. (The Teacher's Corner: A Note on the Correlation of Parts with Wholes. Am Stat 22, 41-41, 1968) should be cited, as they are the first to produce such a figure for the correlation between change and the baseline. Another paper of interest is the one by Sutherland, T. M. (The correlation between feed efficiency and rate of gain, a ratio and its denominator. Biometrics 21, 739-749, 1965). Oldham's method is to address this issue of wrong null hypothesis by testing the difference in the variances of X and Y. Other methods are also available as discussed in my previous paper (Tu and Gilthorpe 2007) cited by the authors. As discussed by the authors, Oldham's method or any others based on the same idea has its limitation, if data may be truncated. A possible solution is to design a more sensitive tool to measure the outcome.

We appreciate you noting these papers, and have added a citation to the first visualization of the surface we present in Figure 1. We also continue to emphasize prior work by Tu and coauthors that introduce methods to appropriately test null hypotheses in the presence of mathematical coupling.

We also agree, with the reviewer and with others working in this area, that more sensitive tools may alleviate concerns due to truncation and the resulting ceiling effects. Indeed, our prior work using kinematic data developed an average scaled distance from the center of a healthy control population (Cortes et al., 2017). In some sense, though, there is an inherent upper limit on recovery – meaning that the insights in our manuscript will be relevant even when better measures of function are available.

3. For the prediction, I think we need to be cautious to use R2 for comparing the performance of different studies. Because R2 is the square of r, i.e. the correlation coefficient. Consequently, the range of R2 is also restricted by the same conditions as r is. Therefore, if different studies have different correlations between X and Y, I do not think their R2 are directly comparable.

We agree with this point completely. In our first submission, we made the connection between R^2^ and correlations explicit for exactly this reason. Because R^2^ can be difficult to interpret, and indeed has been the focus of confusion and misinterpretation in the past, we have urged caution in using R^2^ but deferred this section to an appendix, and removed R^2^ from our reported results.

4. Moreover, a high R2 does not necessarily mean that patients' recovery can be precisely predicted. Suppose a zero correlation between X and Y, the R2 for using X to predict Y – X will be 0.71^2 = 0.5. Although half of the change's variance can be predicted by the model (i.e. by the baseline value X), this model is useless. This is because R2=0.5 is what we would expect from two sets of random numbers. The baseline and the follow-up values behave like two random variables with zero correlation, this means that X has no use for predicting Y.

We agree here as well – in fact, this example has been used to illustrate why reported strong correlations between baseline and change may not mean that recovery is predictable from baseline values. We are concerned, though, that this correct observation has been taken too far in some cases: while a high R^2^ doesn’t guarantee that recovery can be predicted, it is possible that recovery can be predicted well when R^2^ is high.

As in the previous comment, we think it is important to remain more focused on correlations and variance ratios as measures of association, and prediction error as a measure of prediction accuracy. We have therefore moved most of our discussion of R^2^ to the supplement.

5. Regarding to subgroups of patients with different responses, the authors are correct in giving warnings on identifying clusters of patients by using unsupervised methods. Due to regression to the mean, patients with greater baseline diseases are more likely to be identified as good responders, while those with milder diseases are more likely to be identified as poor responders. In fact, the response is actually a continuous spectrum.

The inappropriate use of unsupervised methods can suggest clusters that do not in fact exist in the data. As discussed in Hawe et al. (2018), using the results of an inappropriate clustering analysis can misleadingly suggest the presence of proportional recovery. The focus in that paper was on “random” recovery but, as you note, regression to the mean is an important phenomenon that could drive similar results. We have added this to our discussion of clustering.

We think it is important to reiterate warnings about the inappropriate use of unsupervised methods. A major contribution of our work, we think, is the development of an inferential procedure to test against the specific null hypothesis considered in Hawe et al. (2018). While “random” recovery and regression to the mean can give the illusion of proportional recovery after inappropriate clustering, our results suggest that in at least some dataset there are biologically meaningful subgroups with different recovery patterns. Confirming the existing of these groups, finding ways to accurately classify patients early, and developing targeted therapies to promote recovery is important future work.

6. Finally, I feel a little uneasy that equating the relation between change and the baseline values to the proportional recovery rule. I feel the latter is more akin to the percentage change. Please see the paper: Testing the relation between percentage change and baseline value. Sci Rep 6, 23247 (2016).

Thank you for pointing out this useful paper, and we can see where confusion can arise. The proportional recovery rule is a regression of change (y-x) on initial impairment (66-x), where 66 is the maximum value for the outcome measure and is presumed to be 66 for all patients before stroke. The regression does not include an intercept, and the slope on initial impairment is typically interpreted as the proportion of function lost due to stroke that is expected to be regained during recovery. Put differently, this is the expected change in the amount of recovery for each one unit change in initial impairment. Thus, the slope is interpreted as a proportion or percent – but the regression uses change as an outcome rather than modeling the percent γ−xX as a function of the baseline.

I think the authors made great efforts to clarify various misconceptions about testing the relation between change and the baseline. To get your messages across the intended readers, I suggest making your discussion more focused. For example, I do not feel that introducing bootstrapping or GAM is necessary. In contrast, the key concepts of mathematical coupling and regression to the mean were not explained in the paper. In my experience, they are among the most different concepts to comprehend even for statisticians.

Thank you for the positive assessment of our work, and your suggestions on how it can be improved. As we discussed in our response to your first comment, we have worked to provide the background necessary to synthesize a wide-ranging debate and provide novel insights and contributions. We have chosen to retain the GAM model and the bootstrap in our revisions: the GAM can smoothly capture non-linear associations when they exist, and the bootstrap is a now common inferential procedure. The technical details of the GAM are deferred to a simulation, and we have tried to explain the bootstrap in an accessible way. Meanwhile, in light of your suggestion to make our discussion more focused, we have revised other elements of our paper. Most notably, we have shifted our consideration of R^2^ to the supplement, and have trimmed text for succinctness where possible. We have also provided a clearer definition of coupling and regression to the mean to frame our discussions.

Reviewer #3 (Recommendations for the authors):This is a defence of the Proportional Recovery Rule (PRR), which asserts that stroke survivors recover a fixed proportion of lost function after stroke, from recent, formal/statistical inspired criticisms. The analyses and data show that the PRR is likely to be relevant to ongoing efforts to understand and model post-stroke recovery, and the initial severity of upper limb motor impairments (hemiparesis) after stroke predicts 35-55% of the variance in their subsequent recovery from those deficits.

We appreciate the time taken to evaluate our manuscript. We were motivated by recent well-founded statistical critiques of the PRR, which rightly pointed out issues that can lead to misleading effect sizes, exaggerated claims of statistical significance, or challenges in distinguishing proportional recovery from other recovery mechanisms. Our goal, discussed in detail below, was to synthesize and contribute new insights to this ongoing statistical debate; we then compare several models for recovery in using a prediction-oriented measure of model performance, and conclude that the PRR is likely to be biologically and statistically relevant in models for poststroke recovery moving forward.

This manuscript includes a lot of careful analyses that seem reasonable to me and are novel as far as I know in this specific domain. The authors' conclusions also appear to be supported by their methods and data – and their data are impressive, with three large, relevant datasets. I note that other reviewers have identified some detailed issues with the analysis and the text, but in the main the authors have done a very good job of addressing those concerns. The one exception, in my view at least, concerns the best strategy for dealing with patients at or near ceiling in the first two weeks post-stroke. This issue was raised by another reviewer, with whom I co-authored a paper outlining our favoured response, which is to exclude these patients from analyses seeking to quantify the explanatory power of the PRR (Bonkhoff et al., 2021, JNNP). By failing to exclude those acute-near-ceiling patients, the authors have left me somewhat sceptical of the reliability of the variances explained that they report. But this is perhaps more a matter of taste than of statistical rigour, and I can certainly appreciate the authors' reluctance to exclude hard-won empirical data. In their place, I might reference this issue as a possible limitation in the manuscript.In other words, this paper is not 'broken' in my view, so I have no objection to its publication – and no real requirements for revisions.

Critiques of the PRR and subsequent analyses have been statistically rigorous; we appreciate the reviewer’s comment that our work here is consistent with the care shown in these studies, and that our conclusions are supported by the methods and results.

We apologize for overlooking the suggestion for dealing with ceiling effects raised in the first round of review, and for not responding more fully at that time. The strategy presented is to exclude patients whose baseline scores are above a predetermined threshold. Because this group typically has lower outcome variance at follow-up, the reviewer is correct that omitting these patients would be expected to decrease single-subject prediction accuracy and the proportion of outcome variance explained.

We agree that how best to handle patients at or near ceiling is a matter of preference and analysis goals. In the data we consider, many or most patients have baseline FM scores >= 45. This is a large patient population, and understanding recovery in this group is important scientifically; we hesitate to discard a substantial fraction of our data and miss the opportunity to model recovery for mild to moderate affected patients at baseline. A possible strategy for future work is to incorporate the Tobit-like structure we implemented to evaluate constant recovery into other models for recovery. In any case, it is important to recognize that the narrower outcome distribution for patients at or near ceiling at baseline will affect measures of model performance. We have made this limitation more clear in our revised manuscript.

That said, I also do not believe that this paper makes a particularly compelling contribution to the field: it 'defeats' a straw man caricature of the criticisms made of the PRR, and offers evidence in support of a position with which even the rule's most critical commentators already agree (and indeed for which some of them have already published supporting evidence).The original criticism of the PRR was that the analyses used to support it would yield apparently enormous effect sizes entirely regardless of whether the PRR was really relevant to recovery (Hope et al., 2019, Brain). There was never any assertion that the PRR was irrelevant to recovery: that conclusion could never have been justified merely from the recognition that the empirical support for the PRR was unreliable. A follow-up analysis with a large dataset (albeit not as large as that used here) suggested that the PRR was indeed really relevant to recovery, but that it explained a less of the variance in recovery than prior empirical analyses had suggested (Bonkhoff et al., 2020, Brain). That latter work shares much in common with the analyses presented here: i.e., it is a model comparison analysis, with the PRR as one of the considered models. The current work is also a model comparison analysis, with the PRR as one of the considered models, but implemented with different methods and tested against a lot more data. In other words, this paper is in my view a conceptual replication of the earlier study: using different methods and data to run a broadly similar analysis, which yields broadly similar conclusions to those reported previously. Similarly, while the lengthy discussion of clustering is well done, it adds little (in my view) to the conclusions already drawn in Bonkhoff et al., 2022, JNNP.In other words, much of what this manuscript claims to prove, in response to the critics, has already been reported by some of those same critics. In this sense, the paper is akin to a conceptual replication; using excellent data and novel methods (for the domain at least) to draw conclusions that converge with what has come before, and expressing it all in an accessible manner. These are all strengths in my view: I would merely ask that the links to prior results be made more explicit, at least so that others can follow the timeline of the debate more easily. Or indeed if it's the authors' view that I am wrong, that this be justified more explicitly.

Broad points of agreement around the proportional recovery rule have indeed emerged, and some of our results are similar to those reported by critics of the PRR in recent work. That said, our goal is not to defeat a straw man argument against the PRR, and the contributions of this manuscript go beyond a conceptual replication of previous work.

Critiques of the PRR were careful and limited in their conclusions. For example, Hope et al. (2019) say in their discussion that they “are not claiming that the proportional recovery rule is wrong”, only that they do not see evidence that the rule holds and that other models for recovery are possible. But the title of that paper is “Recovery after stroke: not so proportional after all?”; Hawe et al. (2018) included some similar caveats in the discussion of a paper titled “Taking proportional out of recovery.” Senesh and Reinkensmeyer (2019) (title: “Breaking Proportional Recovery After Stroke”) summarized these papers in their abstract by saying they “explained that [proportional recovery] should be expected because of mathematical coupling between the baseline and change score”. Authors may not have asserted the PRR was irrelevant for recovery, but readers could be forgiven for thinking they had.

Those critiques and more recent work contain important points that we agree with: that early top-line effect sizes were overly optimistic due to counter-intuitive statistical issues; that proportional recovery is useful for understanding recovery, although it explains less outcome variance than one might hope for single-subject prediction; and that different subpopulations exhibit different recovery outcomes. Within those points of agreement, interpretations of results and points of emphasis can differ. For instance, Bonkhoff et al. (2020) summarizes some analysis results by noting that the “first model comparison … pointed to a combination of proportion to lost function and constant recovery” among fitters; our view of the same results is that standard regression and proportional recovery are virtually indistinguishable, and both far outperform constant or proportional to spared models for recovery. Both statements are true, and yet highlighting that different conclusions can be drawn may be useful to readers of this literature.

If the present manuscript contained only different interpretations of past results and analyses of new datasets, it might be considered a conceptual replication of other work. Our manuscript is broader than that, however. We synthesize a broad collection of critiques, draw new statistical insights, and examine those insights in previously reported datasets. Our discussion of correlations as measures of association, for example, illustrates that statistical evidence for non-artifactual associations can be obtained even in the “degenerate” region described in Hope et al. (2019). Several papers have highlighted the difficulty in obtaining accurate null distributions in the presence of coupling and ceiling effects; we have made specific proposals in response. And our work on clustering does indeed differ from and add to the conclusions in Bonkhoff et al. (2022). That paper thresholds initial impairment to separate severe and non-severe cases, and uses Bayesian hierarchical models and model selection approaches to understand whether these groups have different recovery patterns. Our point, meanwhile, is that the severely affected population contains biologically meaningful subgroups – recoverers and non-recoverers – and that these can be identified using appropriate unsupervised learning methods. Hawe et al. (2018) had noted that inappropriately clustering when recovery is random can produce results that mimic proportional recovery; one of our contributions is a hypothesis test for the null of “random recovery” and evidence supporting the separation of recoverers and non-recoverers (rather than severe and non-severe patients at baseline).

Critics of the PRR introduced a range of valid statistical concerns and have produced results that refine the field’s understanding of recovery. Our work attempts to acknowledge those contributions, and in our revisions we have tried to more explicitly discuss the timeline of this debate and highlight prior findings. We have also tried to be more clear about the ways in which our work goes beyond a conceptual replication of past work.

Finally, I am concerned that this issue is rather too narrow to appeal to the readership of eLife. To bolster the more general appeal, I would recommend adding a paragraph or two on variants of this debate that have raged in other subjects – examples of which the authors themselves have given in other papers on this topic.

Thank you for raising this point. While our primary motivation is the appropriate analysis of data that arise in the context of stroke recovery, the same problems are and will continue to be present across a wide range of domains. We agree that being clear about this will broaden the appeal of this paper, and have made this point in the abstract and introduction.

References

Bonkhoff, Anna K, Thomas Hope, Danilo Bzdok, Adrian G Guggisberg, Rachel L Hawe, Sean P Dukelow, Anne K Rehme, Gereon R Fink, Christian Grefkes, and Howard Bowman. 2020. “Bringing proportional recovery into proportion: Bayesian modelling of post-stroke motor impairment.” Brain.

Bonkhoff, A. K., Hope, T., Bzdok, D., Guggisberg, A. G., Hawe, R. L., Dukelow, S. P., … and Bowman, H. (2022). Recovery after stroke: the severely impaired are a distinct group. Journal of Neurology, Neurosurgery and Psychiatry, 93(4), 369-378.

Cortes, J. C., Goldsmith, J., Harran, M. D., Xu, J., Kim, N., Schambra, H. M., … and Kitago, T. (2017). A short and distinct time window for recovery of arm motor control early after stroke revealed with a global measure of trajectory kinematics. Neurorehabilitation and neural repair, 31(6), 552-560.

Hawe, Rachel L, Stephen H Scott, and Sean P Dukelow. 2019. “Taking Proportional Out of Stroke Recovery.” Stroke 50 (1): 204–11.

Hope, Thomas M H, Karl Friston, Cathy J Price, Alex P Leff, Pia Rotshtein, and Howard Bowman. 2018. “Recovery after stroke: not so proportional after all?” Brain 142 (1): 15–22.

Senesh, Merav R., and David J. Reinkensmeyer. "Breaking proportional recovery after stroke." Neurorehabilitation and neural repair 33.11 (2019): 888-901.

Tu, Yu-Kang. "Testing the relation between percentage change and baseline value." Scientific reports 6.1 (2016): 1-8.